# Integral Imprecise Probability Metrics

**Siu Lun Chau**
Nanyang Technological University
Singapore

**Michele Caprio**
University of Manchester
United Kingdom

**Krikamol Muandet**
RI Lab, CISPA
Saarbrücken, Germany

## Abstract

Quantifying differences between probability distributions is fundamental to statistics and machine learning, primarily for comparing statistical uncertainty. In contrast, epistemic uncertainty—due to incomplete knowledge—requires richer representations than those offered by classical probability. Imprecise probability (IP) theory offers such models, capturing ambiguity and partial belief. This has driven growing interest in imprecise probabilistic machine learning (IPML), where inference and decision-making rely on broader uncertainty models—highlighting the need for metrics beyond classical probability. This work introduces the Integral Imprecise Probability Metric (IIPM) framework, a Choquet integral-based generalisation of classical Integral Probability Metrics (IPMs) to the setting of capacities—a broad class of IP models encompassing many existing ones, including lower probabilities, probability intervals, belief functions, and more. Theoretically, we establish conditions under which IIPM serves as a valid metric and metrises a form of weak convergence of capacities. Practically, IIPM not only enables comparison across different IP models but also supports the quantification of epistemic uncertainty (EU) within a single IP model. In particular, by comparing an IP model with its conjugate, IIPM gives rise to a new class of EU measures—Maximum Mean Imprecisions (MMIs)—which satisfy key axiomatic properties proposed in the uncertainty quantification literature. We validate MMI through selective classification experiments, demonstrating strong empirical performance against established EU measures, and outperforming them when classical methods struggle to scale to a large number of classes. Our work advances both theory and practice in IPML, offering a principled framework for comparing and quantifying epistemic uncertainty under imprecision.

## 1 Introduction

Inductive reasoning underpins artificial intelligence, sciences, and human judgment by enabling generalisation from observed evidence to unobserved phenomena. This requires navigating effectively the boundary between certain and uncertain information. Measure-theoretic probability theory, formalised by Kolmogorov [1], provides the mathematical foundation widely adopted to represent and manage such uncertainties. In machine learning (ML), its use is pervasive—from modelling data-generating processes to guiding predictions and decision-making. Probability theory primarily addresses uncertainties described as 'risk', 'first-order uncertainty', 'statistical uncertainty', or 'known unknowns' [2], commonly referred to as *aleatoric uncertainty* in ML [3, 4].

A growing focus in ML concerns another form of uncertainty arising from ignorance or lack of knowledge—termed 'Knightian uncertainty' [5], 'second-order uncertainty', or 'unknown unknowns', commonly known as *epistemic uncertainty (EU)*. While probability theory adequately models aleatoric uncertainty, scholars [6–8] argue that classical *precise* probability, relying on a single probability measure, falls short in representing EU. It struggles to formally model missing, uncertain, or qualitative data [9, Section 1.3] and fails to distinguish between indifference (equal belief) and

39th Conference on Neural Information Processing Systems (NeurIPS 2025).

genuine ignorance (absence of knowledge) [8, Section 1.1.4]; see also Hájek's [10] discussion on Laplace's and Bertrand's paradox [11]. At a more fundamental level, when probability is used to encode subjective belief or confidence, the additivity axiom of Kolmogorov implies that high uncertainty (low confidence) about an event necessarily entails high certainty (high confidence) about its complement. Yet under limited information, this may not hold—we might reasonably remain ambiguous about both. These limitations contribute to practical issues in ML, including overconfidence [12], poor generalisation [13, 14], and inadequate handling of partial ignorance [15]. See Manchingal and Cuzzolin [16] for a recent discussion on the need to equip machine learning systems with epistemic uncertainty awareness.

Uncertainty-centric ML paradigms have emerged to address the limitations of classical approaches by explicitly modelling EU. Examples include Bayesian ML [17] and evidential learning [18] —each grounded in distinct philosophies with respective strengths and limitations. A growing alternative, known as Imprecise Probabilistic Machine Learning (IPML), focuses on inference and decision-making using generalisations of classical probability, commonly known as *imprecise probabilities (IPs)* [8, 19, 20]. Examples of IP models include interval probabilities [6], random sets [21], fuzzy measures [22], belief functions [23], higher-order probabilities [24, 25], possibility theory [26], credal sets [27, 28], and lower/upper probabilities [8, 29]. Many of these fall under the broader class of non-additive set functions called *Choquet capacities* [30]; see Definition 2 and Figure 2 in Appendix A for their generality. IPML provides principled tools to handle imprecision arising from model misspecification and data uncertainty, with growing applications in classification [31–33], hypothesis testing [34, 35], scoring rules [36, 37], conformal prediction [38, 39], computer vision [40, 41], probabilistic programming [42], explainability [43–46], neural networks [31, 47, 48], learning theory [49], causal inference [50, 51], active and continual learning [52, 53], fixed point theory [54], and many more.

Motivated by these advances, this work contributes to the core development of IPML by introducing a first-of-its-kind family of statistical distances for imprecise probabilities, termed *Integral Imprecise Probability Metric (IIPM)*. IIPMs strictly generalise the classical Integral Probability Metric (IPM) framework [55] by leveraging Choquet integration [30] to compare capacities, thereby recovering IPMs as a special case. With appropriate choices of function classes, IIPMs extend well-known metrics—such as the Dudley metric and total variation distance—to the setting of IPs. Theoretically, we establish conditions under which IIPMs serve as valid metrics and metrise the weak convergence of capacities. As an illustration, we show how IIPMs recover key theoretical results in a recent optimal transport under $\epsilon$-contamination [56] problem. Beyond these theoretical contributions, IIPMs naturally induce a new class of EU measures by quantifying the discrepancy between a capacity and its conjugate (see Section 2.1). This yields the *Maximum Mean Imprecision (MMI)*, a family of epistemic uncertainty quantification (UQ) measures that satisfy several foundational axioms from the literature, along with an efficiently computable upper bound. We empirically validate MMI and its linear-time upper bound in selective classification tasks [57], demonstrating strong empirical performance and improved scalability over existing epistemic UQ measures—particularly in settings with a large number of classes, where conventional methods often fail.

The paper is structured as follows. Section 2 reviews IPMs and IP, followed by IIPM and MMI in Section 3 and 4, respectively. Section 5 discusses related work, Section 6 presents empirical results, and Section 7 concludes with discussion. All proofs and derivations can be found in Appendix B.

## 2  Background

**Notations.**  Let $(\mathcal{X}, \Sigma_{\mathcal{X}})$ be our measurable space, where $\mathcal{X}$ is a non-empty set and $\Sigma_{\mathcal{X}}$ a $\sigma$-algebra. We denote $C_b(\mathcal{X})$ as the space of continuous bounded real-valued measurable functions on $\mathcal{X}$. We denote $\mathcal{V}(\mathcal{X})$ the space of capacities $\nu$ on $(\mathcal{X}, \Sigma_{\mathcal{X}})$ (see Definition 2), while $\underline{\mathcal{P}}(\mathcal{X}), \overline{\mathcal{P}}(\mathcal{X})$ represent the spaces of lower and upper probability measures $\underline{P}, \overline{P}$ on $(\mathcal{X}, \Sigma_{\mathcal{X}})$ (see Definition 3) respectively, and $\mathcal{P}(\mathcal{X})$ the space of probability measures $P$ on $(\mathcal{X}, \Sigma_{\mathcal{X}})$.

**Integral Probability Metric.**  Quantifying the difference between two probability measures $P, Q \in \mathcal{P}(\mathcal{X})$ is a ubiquitous task in statistics and machine learning. Popular classes of divergence include the $\phi$-divergence [58, 59], Bregman divergence [60, 61], and $\alpha$-divergence [62]. Another widely used family of divergence measures, central to our paper, is the Integral Probability Metric (IPM) [55, 63], also known as *probability metrics with a $\zeta$-structure* [64]. Given a set of continuous bounded

real-valued measurable functions $\mathcal{F} \subseteq C_b(\mathcal{X})$, and probability measures $P, Q \in \mathcal{P}(\mathcal{X})$, the IPM associated to $\mathcal{F}$ is defined as

$$\mathrm{IPM}_{\mathcal{F}}(P, Q) := \sup_{f \in \mathcal{F}} \left\{ \left| \int f dP - \int f dQ \right| \right\},$$

where $\int f dP$ is the Lebesgue integral of $f$ with respect to $P$. With suitable choices of $\mathcal{F}$, we recover popular probability distances, including the Dudley metric [65], Kantorovich metric and Wasserstein distance [66], total variation distance and Kolmogorov distance [1], and kernel distance commonly known as maximum mean discrepency (MMD) [67]. IPMs have also found numerous theoretical applications—appearing in proofs of central limit theorems [68], empirical process theory [69], and the metrisation of weak topology on $\mathcal{P}(\mathcal{X})$ [70, Chapter 11]—as well as practical applications in hypothesis testing [71] and generative model training [72].

## 2.1 Beyond Precise Probability and Expectations: Capacities and Choquet Integration

While powerful, the IPM framework is limited to probability measures. We extend it to capacities—a broad class encompassing many IP models. This section introduces key concepts: probabilities, capacities, lower probabilities, and Choquet integration.

**Probabilities.** We include here the formal definition of probability measure for completeness,

**Definition 1** (Kolmogorov [1]). *A probability measure $P \in \mathcal{P}(\mathcal{X})$ on a measurable space $(\mathcal{X}, \Sigma_{\mathcal{X}})$ is a set function $P : \Sigma_{\mathcal{X}} \to [0, 1]$ such that (i) $P(A) \geq 0$ for any $A \in \Sigma_{\mathcal{X}}$ (ii) $P(\mathcal{X}) = 1$, and (iii) for any sequence of disjoint sets $\{A_i\}_{i \geq 1}$ with each $A_i \in \Sigma_{\mathcal{X}}$, $P(\cup_{i \geq 1} A_i) = \sum_{i \geq 1} P(A_i)$.*

We adopt countable additivity for generality, though some scholars prefer finite additivity. Kolmogorov's probability theory elegantly applies measure theory to the measurable space $(\mathcal{X}, \Sigma_{\mathcal{X}})$, where the additive measure $P$ assigns credence (degrees of belief) to events in $\Sigma_{\mathcal{X}}$.

**Capacities.** Nonetheless, various scholars have argued that the additivity axiom is too restrictive when dealing with uncertainty involving partial ignorance and ambiguity [73]. This leads to an interest in studying more general non-additive measures, starting with capacities:

**Definition 2** (Choquet [30]). *A capacity $\nu \in \mathcal{V}(\mathcal{X})$ on a measurable space $(\mathcal{X}, \Sigma_{\mathcal{X}})$ is a set function $\nu : \Sigma_{\mathcal{X}} \to [0, 1]$ such that (i) $\nu(\emptyset) = 0, \nu(\mathcal{X}) = 1$, (ii) for any $A, B \in \Sigma_{\mathcal{X}}$ with $A \subseteq B$, $\nu(A) \leq \nu(B)$, and (iii) $\nu$ is continuous from above and below.*

The last condition can be omitted when working with finite spaces $\mathcal{X}$ [74, Section 6.8]. Furthermore, we say a capacity is convex, or 2-monotonic, if for any $A, B \in \Sigma_{\mathcal{X}}$, $\nu(A \cup B) + \nu(A \cap B) \geq \nu(A) + \nu(B)$. Probability measures $P$ are always capacities (and 2-monotonic), but not vice versa. 2-monotonicity equips capacities with more structure to make them computationally easier to work with (see e.g. Lemma 5), while still including as particular cases many of the IP models, such as $\epsilon$-contamination models [75], belief functions [76], and probability intervals [6].

**Lower Probabilities.** Given capacity $\nu$, we define its conjugate $\nu^{\star} : \Sigma_{\mathcal{X}} \to [0, 1]$, as $\nu^{\star}(A) = 1 - \nu(A^c)$, for all $A \in \Sigma_{\mathcal{X}}$. The conjugate capacity serves as a dual representation of the original capacity, meaning one fully determines the other. We clarify its significance and interpretation after introducing lower probability, which lies between capacities and probability measures in generality.

**Definition 3.** *[Cerreia-Vioglio et al. 77, Section 2.1.viii] A set function $\underline{P} : \Sigma_{\mathcal{X}} \to [0, 1]$ is a lower probability if there exists a compact set $\mathcal{C} \subseteq \mathcal{P}(\mathcal{X})$ of probability measures such that $\underline{P}(A) = \min_{P \in \mathcal{C}} P(A)$ for all $A \in \Sigma_{\mathcal{X}}$.*

The compact set $\mathcal{C}$ of measures is called the credal set and the compactness here has to be understood in the weak$^{\star}$ topology [77, P.2 Remark 1]. Lower probabilities are always capacities but not vice versa [77, Section 2.1]. The conjugate of a lower probability $\underline{P}$ is the upper probability $\overline{P}$, defined by $\overline{P}(A) = 1 - \underline{P}(A^c) = \max_{P \in \mathcal{C}} P(A)$ for $A \in \Sigma_{\mathcal{X}}$. In IPML, lower probabilities are typically constructed from a set of probabilistic predictors, e.g., $\mathcal{C} = \{\hat{p}_i(Y \mid X = x)\}_{i=1}^m$ [78–80, 48], or via basic probability assignments [81], as in Dempster–Shafer theory [76], or by constructing predictive intervals [82, 83].

Lower and upper probabilities offer two complementary ways of expressing credence in an event's truthfulness. Specifically, the lower probability $\underline{P}(A)$ reflects the minimal credence assigned directly to event $A$, while the upper probability $\overline{P}(A)$ captures the maximal credence, computed as one minus the lower probability of the complement, i.e., $1 - \underline{P}(A^c)$. These represent, respectively, pessimistic and optimistic assessments of $A$'s objective likelihood. For a subjectivist interpretation of lower and upper probabilities grounded in de Finetti's framework of probability as coherent betting rates, see Walley [8, Section 2.3.5]. In classical probability, these bounds coincide, since $P(A) = 1 - P(A^c)$ for any event $A \in \Sigma_{\mathcal{X}}$. In other words, $P$ is self-conjugate.

**Choquet Integration.**    Another concept central to our contribution is the use of Choquet integrals to measure the discrepancies between capacities, akin to how Lebesgue integrals play a key role for IPM. Choquet integration [30] generalises Lebesgue integration for probability measures to capacities:

**Definition 4** (Choquet Integral)**.** *Given a capacity $\nu \in \mathcal{V}(\mathcal{X})$ and a real-valued function $f$ on $\mathcal{X}$, the Choquet integral of $f$ with respect to $\nu$ is*

$$\oint f d\nu = \int_0^\infty \nu(\{f^+ \geq t\})dt - \int_0^\infty \nu(\{f^- \geq t\}dt, \tag{1}$$

*provided the difference on the right-hand side is well defined, where $f^+ := \max(0, f)$, $f^- := -\min(0, f)$, and $\{f \geq t\} := \{x \in \mathcal{X} : f(x) \geq t\}$.*

When both terms on the right-hand side of Equation (1) are finite, then we say $f$ is Choquet integrable with respect to $\nu$. When $\nu$ is a probability measure, the Choquet integral becomes the standard Lebesgue integral. If $f$ is bounded, then by Troffaes and De Cooman [19, Proposition C.3.ii], $f$ is Choquet integrable with respect to $\nu$ and the integral can be written as,

$$\oint f d\nu = \underline{f} + \int_{\underline{f}}^{\overline{f}} \nu(\{f \geq t\})\, dt,$$

where $\underline{f} := \inf_{x \in \mathcal{X}} f(x)$ and $\overline{f} := \sup_{x \in \mathcal{X}} f(x)$. The Choquet integral provides a generalised notion of "expectation" that accommodates non-additive uncertainty assignments, as commonly encountered in frameworks such as imprecise probabilities, belief functions, and fuzzy measures. For example, the Choquet integral of a lower probability provides a lower bound to the worst-case expectation across the entire credal set, as demonstrated in the following Lemma.

**Lemma 5.** *For lower probability $\underline{P}$ associated to credal set $\mathcal{C}$, we have $\oint f d\underline{P} \leq \inf_{P \in \mathcal{C}} \int f dP$ for any $f \in C_b(\mathcal{X})$. When $\underline{P}$ is 2-monotonic, the inequality becomes an equality.*

## 3   The Integral Imprecise Probability Metric (IIPM) framework

The motivation of IPM follows from a classical result [70, Lemma 9.3.2], stating that two probability measures are equal if and only if their Lebesgue integrals agree for all $f \in C_b(\mathcal{X})$. Before introducing IIPM, we examine whether an analogous result holds for capacities and Choquet integrals. We follow the conditions in Dudley [70, Lemma 9.3.2] and examine metric spaces $(\mathcal{X}, d)$ for some metric $d$.

**Theorem 6.** *Let $(\mathcal{X}, d)$ be a metric space. For any capacities $\nu, \mu \in \mathcal{V}(\mathcal{X})$, we have $\oint f d\nu = \oint f d\mu$ for all $f \in C_b(\mathcal{X})$, if and only if $\nu = \mu$.*

Narukawa [84, Proposition 7] presents a similar result but for bounded non-negative continuous functions under stricter conditions on the capacities (regularity) and different conditions on $\mathcal{X}$ (locally compact Hausdorff). While Theorem 6 characterises equality of capacities via Choquet integrals, the full class $C_b(\mathcal{X})$ is often too rich for practical use. Following the spirit of IPMs, we define the Integral Imprecise Probability Metric *with respect to* a function class $\mathcal{F} \subseteq C_b(\mathcal{X})$ as follows.

**Definition 7** (Integral Imprecise Probability Metric (IIPM))**.** *For function class $\mathcal{F} \subseteq C_b(\mathcal{X})$ and capacities $\nu, \mu \in \mathcal{V}(\mathcal{X})$, the **integral imprecise probability metric** associated with $\mathcal{F}$ between $\nu, \mu$ is*

$$\mathrm{IIPM}_{\mathcal{F}}(\nu, \mu) := \sup_{f \in \mathcal{F}} \left\{ \left| \oint f d\nu - \oint f d\mu \right| \right\}.$$

As we are working with bounded real-valued functions, the Choquet integrals are well-defined.

**Corollary 8.** *For any $P, Q \in \mathcal{P}(\mathcal{X})$ and $\mathcal{F} \subseteq C_b(\mathcal{X})$, $\mathrm{IIPM}_{\mathcal{F}}(P,Q) = \mathrm{IPM}_{\mathcal{F}}(P,Q)$.*

This illustrates that IIPM generalises IPM, but not vice versa, since Lebesgue integrals are defined for probability measures, not capacities. We verify $\mathrm{IIPM}_{\mathcal{F}}$ serves as a pseudometric on $\mathcal{V}(\mathcal{X})$.

**Proposition 9.** *For any $\mathcal{F} \subseteq C_b(\mathcal{X})$, $\mathrm{IIPM}_{\mathcal{F}}$ is a pseudometric on $\mathcal{V}(\mathcal{X})$; it is **non-negative**, **symmetric**, and satisfies the **triangle inequality**.*

When $\mathcal{F}$ is rich enough for $\mathrm{IIPM}_{\mathcal{F}}$ to be point-separating—for instance, when $\mathcal{F} = C_b(\mathcal{X})$—then $\mathrm{IIPM}_{\mathcal{F}}$ defines a proper metric on $\mathcal{F}(\mathcal{X})$. More desirably, albeit stronger, is for convergence in $\mathrm{IIPM}_{\mathcal{F}}$ to imply and necessitate convergence in another established sense, i.e., $\mathrm{IIPM}_{\mathcal{F}}(\mu, \nu) \to 0$ should ideally entail that $\nu$ "truly" converges to $\mu$, where "truly" refers to convergence under some other notion of interest independent to the choice of $\mathrm{IIPM}_{\mathcal{F}}$. If this implication holds, then $\mathrm{IIPM}_{\mathcal{F}}$ is said to metrise that notion of convergence on $\mathcal{V}(\mathcal{X})$. Studying the metrisation properties of divergences is important for analysing the reliability of inference procedures that depend on them [85]. In our case, we follow Feng and Nguyen [86] and consider the Choquet weak convergence of capacities: a sequence $\nu_n \in \mathcal{V}(\mathcal{X})$ is said to converge to $\nu$ in this sense if $\oint f d\nu_n \to \oint f d\nu$ for all $f \in C_b(\mathcal{X})$. This leads to the following natural question: What conditions on $\mathcal{F}$ are sufficient for $\mathrm{IIPM}_{\mathcal{F}}$ to metrise Choquet weak convergence on $\mathcal{V}(\mathcal{X})$? Our initial analysis gives the following answer:

**Theorem 10.** *Let $\mathcal{F} \subseteq C_b(\mathcal{X})$ be dense in $C_b(\mathcal{X})$ with respect to the $\| \cdot \|_\infty$ norm. Then, $\mathrm{IIPM}_{\mathcal{F}}$ metrises the Choquet weak convergence of $\mathcal{V}(\mathcal{X})$.*

To build intuition for Theorem 10, recall from Theorem 6 that $C_b(\mathcal{X})$ is point-separating on $\mathcal{V}(\mathcal{X})$. If $\mathcal{F}$ is dense in $C_b(\mathcal{X})$ under the $\| \cdot \|_\infty$ norm, then this allows us to "import" the separating ability of $C_b(\mathcal{X})$ to $\mathcal{F}$. Our analysis follows from Gretton et al. [71, Theorem 18] for IPMs. However, more general conditions on $\mathcal{F}$ (and $\mathcal{X}$) for metrising weak convergence in $\mathcal{P}(\mathcal{X})$ have been studied—see, e.g., Sriperumbudur et al. [87], Sriperumbudur [88]. Extending such analysis to the case of capacities and understanding the precise conditions under which $\mathcal{F}$ and $\mathcal{X}$ ensure metrisation of Choquet weak convergence on $\mathcal{V}(\mathcal{X})$ is definitely an interesting direction for future work.

### 3.1 Example use cases of the IIPM framework

While general, capacities are often too complex for practical use without additional structure. We therefore focus on lower probabilities (cf. Definition 3) for the remainder of the paper. Nonetheless, our framework also applies to other IP models—such as probability intervals, belief functions, and possibility measures—all of which are structured subclasses of capacities. We show how the IIPM framework gives rise to new divergences and can be used to prove existing results in IPML differently.

**The Lower Dudley Metric.** We start with a lower probability version of the Dudley metric.

**Definition 11** (Lower Dudley Metric). *For a metric space $(\mathcal{X}, d)$, let $\mathcal{F}_d := \{f \in C_b(\mathcal{X}) : \|f\|_{BL} \leq 1\}$, where the Lipschitz norm $\|f\|_{BL} := \|f\|_\infty + \|f\|_L$ and $\|f\|_L := \sup\{|f(x)-f(y)|/d(x,y) : x \neq y \in \mathcal{X}\}$, we define $\mathrm{IIPM}_{\mathcal{F}_d}$ as the Lower Dudley metric.*

**Proposition 12.** *The Lower Dudley metric metrises Choquet weak convergence on $\underline{\mathcal{P}}(\mathcal{X})$.*

This result immediately follows from Theorem 10, and showcases that the lower Dudley metric can be applied in settings where analysing weak convergence of conservative uncertainty assessment is essential, e.g., in robust statistics and safety-critical applications.

**Lower Total Variation.** We can define the lower total variation distance as follows,

**Definition 13** (Lower Total Variation Distances). *Let $\underline{P}, \underline{Q} \in \underline{\mathcal{P}}(\mathcal{X})$ be two lower probabilities on $(\mathcal{X}, \Sigma_{\mathcal{X}})$ and $\mathcal{F}_{TV} := \{\mathbf{1}_A : A \in \Sigma_{\mathcal{X}}\}$, then we define $\mathrm{IIPM}_{\mathcal{F}_{TV}}(\underline{P}, \underline{Q}) = \sup_{A \in \Sigma_{\mathcal{X}}} |\underline{P}(A) - \underline{Q}(A)|$ as the Lower Total Variation Distances (LTV) between $\underline{P}, \underline{Q}$.*

This equality follows from Troffaes and De Cooman [19, Proposition C.5(ii)]. In fact, the LTV distance was also considered in Levin and Peres [89, Proposition 4.2] but was not derived from any integral-based metric framework. It is also important to highlight that many results that hold for IPMs do not carry over to IIPMs due to the loss of additivity. Specifically, while expectations under IPMs are linear, expectations under IIPMs—typically modelled using the Choquet integral—are generally non-linear and, at best, super or sublinear [90]. The following is one such example:

**Remark 14.** *Let $\mathcal{X}$ be finite. For any $P, Q \in \mathcal{P}(\mathcal{X})$, we have $\mathrm{IPM}_{\mathcal{F}_{TV}}(P, Q) = \sup_{A \in \Sigma_{\mathcal{X}}} |P(A) - Q(A)| = \frac{1}{2} \sum_{x \in \mathcal{X}} (P(\{x\}) - Q(\{x\}))$. In contrast, in the imprecise case, there exists $\underline{P}, \underline{Q} \in \underline{\mathcal{P}}(\mathcal{X})$ such that $\mathrm{IIPM}_{\mathcal{F}_{TV}}(\underline{P}, \underline{Q}) := \sup_{A \in \Sigma_{\mathcal{X}}} |\underline{P}(A) - \underline{Q}(A)| \neq \frac{1}{2} \sum_{x \in \mathcal{X}} (\underline{P}(\{x\}) - \underline{Q}(\{x\}))$.*

In subsequent applications of IIPM to epistemic UQ, we adopt the LTV distance for its simplicity.

**Lower Kantorovich problem with $\epsilon$-contamination sets.** To illustrate the theoretical utility of IIPM, we consider the well-known $\epsilon$-contamination model from robust statistics [75]. We show how IIPM rederives a key result from Caprio [56, Theorem 11], namely that the solution to the *restricted lower probability Kantorovich problem* (RLPK) [56, Definition 10] coincides with the classical Kantorovich problem under $\epsilon$-contamination. For completeness, a detailed treatment of the RLPK problem is provided in Appendix D.1. Recall the $\epsilon$-contamination set $\mathcal{P}_{\epsilon,P}$ of a probability measure $P \in \mathcal{P}(\mathcal{X})$ is given by,

$$\mathcal{P}_{\epsilon,P} := \{\tilde{P} \in \mathcal{P}(\mathcal{X}) : \exists R \in \mathcal{P}(\mathcal{X}) \text{ s.t. } \tilde{P}(A) = (1 - \epsilon)P(A) + \epsilon R(A), \forall A \in \Sigma_{\mathcal{X}}\}, \qquad (2)$$

with $\epsilon \in [0, 1]$ the contamination rate of the model. The lower probability corresponding to $\mathcal{P}_{\epsilon,P}$ is derived in Walley [8, Section 2.9.2].

**Lemma 15.** *Let $\mathcal{P}_{\epsilon,P}$ be an $\epsilon$-contaminated model defined in Equation (2). Then, the associated lower probability $\underline{P}_\epsilon$ is given by*

$$\underline{P}_\epsilon(A) = \inf_{\tilde{P} \in \mathcal{P}_{\epsilon,P}} \tilde{P}(A) = \begin{cases} (1 - \epsilon)P(A) & \text{for all } A \in \Sigma_{\mathcal{X}} \setminus \{\mathcal{X}\} \\ 1, & \text{for } A = \mathcal{X} \end{cases}$$

Now we can recover the results from Caprio [56] using the IIPM framework.

**Theorem 16.** *Let $\mathcal{F}_W := \{f \in C_b(\mathcal{X}) : \|f\|_L \leq 1\}$ where $\|f\|_L := \sup_{x,y \in \mathcal{X}} \{|f(x) - f(y)|/c(x,y)\}$, and $c$ the transportation cost in a restricted lower probability Kantorovich (RLPK) problem [56, Definition 10]. Let $\underline{P}_\epsilon$, $\underline{Q}_\epsilon$ be lower probabilities of the $\epsilon$-contaminated models $\mathcal{P}_{\epsilon,P}$ and $\mathcal{P}_{\epsilon,Q}$. Then, $\mathrm{IIPM}_{\mathcal{F}_W}(\underline{P}, \underline{Q})$ coincides with the objective of the RLPK problem, and thus coincides with the classical Kantorovich's optimal transport problem involving $P$ and $Q$.*

We include this result to illustrate the utility and flexibility of the IIPM framework for theoretical analysis. We also highlight that, in general, a version of the Kantorovich-Rubinstein duality theorem for RLPK does not (yet) exist, therefore, we cannot simply associate $\mathrm{IIPM}_{\mathcal{F}_W}$ as the *lower Wasserstein-1 distance*. In Appendix D.2, we continue our analysis on $\epsilon$-contamination models with IIPMs, but using kernels, following the derivation of kernel distances in Gretton et al. [71]. Specifically, we consider $\mathcal{F}_k := \{f \in \mathcal{H}_k : \|f\|_{\mathcal{H}_k} \leq 1\}$, where $k$ is a kernel and $\mathcal{H}_k$ the corresponding reproducing kernel Hilbert space (RKHS). Non-parametric estimator of $\mathrm{IIPM}_{\mathcal{F}_k}(\underline{P}_\epsilon, \underline{Q}_\delta)$ (different contamination level is allowed) is also derived. We defer this result due to space constraints.

## 4 Measuring epistemic uncertainty with IIPM: *Maximum Mean Imprecision*

Beyond its appeal for theoretical analyses, the IIPM framework can find practical application in epistemic UQ for ML. By quantifying the gap between capacities and their conjugates, IIPM captures the divergence between optimistic and pessimistic assessments—giving rise to a new class of EU measures, which we call *Maximum Mean Imprecision (MMI)*. As in Section 3, we illustrate MMI using lower probabilities, though the framework remains broadly applicable to other IP models.

**Definition 17** (Maximum Mean Imprecision). *Let $\mathcal{F} \subseteq C_b(\mathcal{X})$. The **Maximum Mean Imprecision** with respect to $\mathcal{F}$ is the function $\mathrm{MMI}_{\mathcal{F}} : \underline{\mathcal{P}}(\mathcal{X}) \to \mathbb{R}$ defined as,*

$$\mathrm{MMI}_{\mathcal{F}}(\underline{P}) := \mathrm{IIPM}_{\mathcal{F}}(\overline{P}, \underline{P}) = \sup_{f \in \mathcal{F}} \left\{ \oint f d\overline{P} - \oint f d\underline{P} \right\}.$$

Note that we have omitted the absolute value sign between the Choquet integrals. This is intentional: since the upper probability $\overline{P}$ setwise-dominates the lower probability $\underline{P}$, the difference is always non-negative, rendering the absolute value sign unnecessary. To explain MMI's underlying mechanism, we present an alternative formulation of $\mathrm{MMI}_{\mathcal{F}}$.

**Proposition 18.** *The definition of MMI is equivalent to*

$$\mathrm{MMI}_{\mathcal{F}}(\underline{P}) = \sup_{f \in \mathcal{F}} \int_{\underline{f}}^{\overline{f}} 1 - \Big( \underline{P}(\{f < t\}) + \underline{P}(\{f \geq t\}) \Big) dt \tag{3}$$

First, notice that for any $t \in [\underline{f}, \overline{f}]$, $\{f < t\} \cup \{f \geq t\} = \mathcal{X}$. Equation (3) shows that $\mathrm{MMI}_{\mathcal{F}}$ quantifies EU as the largest possible accumulation of disagreement between $1 = \underline{P}(\mathcal{X})$ and the sum of the lower probabilities assigned to the complementary events $\{f < t\}$ and $\{f \geq t\}$ under $\underline{P}$ as we "slide" the threshold $t$. As an illustrative example, we now demonstrate how to quantify the EU for the $\epsilon$-contamination model, previously introduced in Section 3.

**Proposition 19** (MMI on $\epsilon$-contamination set.)**.** *Let $\underline{P}_{\epsilon}$ be the lower probability associated with $\mathcal{P}_{\epsilon,P}$ and $\mathcal{F} \subseteq C_b(\mathcal{X})$. Then $\mathrm{MMI}_{\mathcal{F}}(\underline{P}_{\epsilon}) = \epsilon \left( \sup_{f \in \mathcal{F}} \sup_{x,y \in \mathcal{X}} |f(x) - f(y)| \right)$. For the LTV distance with $\mathcal{F}_{TV} := \{\mathbf{1}_A : A \in \Sigma_{\mathcal{X}}\}$, we have $\mathrm{MMI}_{\mathcal{F}_{TV}}(\underline{P}_{\epsilon}) = \sup_{A \in \Sigma_{\mathcal{X}}} \{\overline{P}_{\epsilon}(A) - \underline{P}_{\epsilon}(A)\} = \epsilon$.*

This aligns with the intuition that contaminating a probability assessment by $\epsilon$ amount should proportionally increase the EU by $\epsilon$. The "unit" of this uncertainty is determined by the variability of the chosen function class. For TV distance, the unit is precisely 1. This result is useful later when we derive an upper bound for the $\mathrm{MMI}_{\mathcal{F}_{TV}}$ of any lower probability in Proposition 21.

**Desirable axiomatic properties for epistemic UQ measure.** While $\mathrm{MMI}_{\mathcal{F}}$ appears to be an intuitive measure of uncertainty, it is important to examine whether it satisfies key axiomatic properties proposed in the literature on credal UQ [91], where the EU modelled using IPs is measured. This is especially relevant given the abundance of UQ measures [4, 92, 93], many of which lack rigorous axiomatic support. The axioms we present below were originally formulated for credal sets and are adapted here for lower probabilities, as both representations mostly convey interchangeable semantics. In particular, Abellán and Klir [94], Jiroušek and Shenoy [95], Hüllermeier et al. [96], and Sale et al. [97] propose that a valid credal uncertainty measure $U : \underline{\mathcal{P}}(\mathcal{X}) \to \mathbb{R}$ should satisfy:

**A1 Non-negativity and boundedness**: (a) $U(\underline{P}) \geq 0$, for all $\underline{P} \in \underline{\mathcal{P}}(\mathcal{X})$; (b) there exists $u \in \mathbb{R}$ such that $U(\underline{P}) \leq u$, for all $\underline{P} \in \underline{\mathcal{P}}(\mathcal{X})$.

**A2 Continuity**: $U$ is a continuous functional.

**A3 Monotonicity**: for $\underline{P}, \underline{Q} \in \underline{\mathcal{P}}(\mathcal{X})$, if $\underline{P}(A) \leq \underline{Q}(A)$ for all $A \in \Sigma_{\mathcal{X}}$, then $U(\underline{P}) \geq U(\underline{Q})$.

**A4 Probability consistency**: for all $P \in P(\mathcal{X})$ we have $U(P) = 0$.

**A5 Sub-additivity**: Let $(\mathcal{X}_1, \Sigma_{\mathcal{X}_1})$ and $(\mathcal{X}_2, \Sigma_{\mathcal{X}_2})$ be measurable spaces and such that $\mathcal{X} = \mathcal{X}_1 \times \mathcal{X}_2$. Let $\underline{P}_1(\cdot) = \underline{P}(\cdot, \mathcal{X}_2)$ and $\underline{P}_2(\cdot) := \underline{P}(\mathcal{X}_1, \cdot)$ be the corresponding marginal lower probabilities on $\mathcal{X}_1$ and $\mathcal{X}_2$ respectively. Then, we have $U(\underline{P}) \leq U(\underline{P}_1) + U(\underline{P}_2)$.

**A6 Additivity**: Following from A5, if $\underline{P}_1$ and $\underline{P}_2$ are independent, then $U(\underline{P}) = U(\underline{P}_1) + U(\underline{P}_2)$.

Note Axiom A6 assumes a notion of independence for IPs, for which multiple, non-equivalent definitions exist; see Couso et al. [98], Cozman [99] for a review. Axioms A1 and A2 set out the basic requirements for U to be "sensible". Axiom A3 states that if a representation $\underline{P}$ is uniformly more conservative than $\underline{Q}$—assigning lower values to all events—then $\underline{P}$ should be deemed more epistemically uncertain. Axiom A4 ensures that precise probabilities correspond to zero EU. Some work [100] instead prefers the measure to reflect pure aleatoric uncertainty when EU is absent. Axiom A5 captures the intuition that joint reasoning reduces uncertainty, as dependencies between events provide additional structure. Axiom A6 complements this by stating that, when no such dependencies are present, joint or separate consideration should not affect the uncertainty measure. A5 and A6 are less relevant when quantifying predictive uncertainty in classification or real-valued regression tasks, since the output spaces are generally not product spaces.

At this point, one may naturally ask whether $\mathrm{MMI}_{\mathcal{F}}$ satisfies these axioms—or under what conditions on $\mathcal{F}$ this can be ensured. Particular care is required for Axioms A5 and A6, which involve reasoning over product spaces. To meaningfully evaluate these axioms, we must clarify the function classes defined over the marginal domains $\mathcal{X}_1$ and $\mathcal{X}_2$. Suppose $\mathcal{X} = \mathcal{X}_1 \times \mathcal{X}_2$ as specified in axiom A5. Let $\mathcal{F}_1 \subseteq C_b(\mathcal{X}_1)$, $\mathcal{F}_2 \subseteq C_b(\mathcal{X}_2)$, and $\mathcal{F}_{12} := \{f \in C_b(\mathcal{X}) \text{ s.t. } f(x_1, x_2) = f_1(x_1) + f_2(x_2), \text{ for some } f_1 \in \mathcal{F}_1, f_2 \in \mathcal{F}_2\}$.

**Theorem 20.** *For any $\mathcal{F} \subseteq C_b(\mathcal{X})$, $\mathrm{MMI}_{\mathcal{F}}$ satisfies axioms **A1-A4**. If $\underline{P}$ is 2-monotonic and with $\mathcal{F} = \mathcal{F}_{12}$ defined above, then $\mathrm{MMI}_{\mathcal{F}_{12}}$ satisfies axioms **A1-A5**. For **A5**, the subadditivity becomes $\mathrm{MMI}_{\mathcal{F}_{12}} \leq \mathrm{MMI}_{\mathcal{F}_1} + \mathrm{MMI}_{\mathcal{F}_2}$. If the notion of independence in A6 is taken to be strong independence in the sense of Cozman [99], **A6** also holds, and $\mathrm{MMI}_{\mathcal{F}_{12}} = \mathrm{MMI}_{\mathcal{F}_1} + \mathrm{MMI}_{\mathcal{F}_2}$.*

Theorem 20 establishes $\mathrm{MMI}_{\mathcal{F}}$ as a principled measure of EU for lower probabilities. Axioms A1–A4 hold for any function class $\mathcal{F} \subseteq C_b(\mathcal{X})$, while A5 and A6 require additional 2-monotonicity—satisfied, for instance, by belief functions and suitably chosen product function spaces in multivariate settings. Examples of credal EU measures that satisfy some of the axioms include: the difference between maximal and minimal entropy within a credal set [100], which fails A3; the generalised Hartley (GH) measure [94], which satisfies all 6 axioms; and the volume of the credal set, which fails A6. Nonetheless, Sale et al. [97] demonstrated that credal set volume fails to be a good EU measure in $K$-class classification problems when $K > 2$ due to the lack of robustness in computation as the dimension $K$ increases. We, therefore, exclude it from our experimental comparisons. We also note that, when $|\mathcal{X}| = 2$ as in binary classification, $\mathrm{MMI}_{\mathcal{F}_{TV}}$ corresponds to the *credal epistemic measure*, $\overline{P}(\{x\}) - \underline{P}(\{x\})$ for $x \in \mathcal{X}$, considered in Hüllermeier et al. [101].

**A linear-time upper bound.** Computing $\mathrm{MMI}_{\mathcal{F}_{TV}}(\underline{P}) = \sup_{A \subseteq \mathcal{X}}\{\overline{P}(A) - \underline{P}(A)\}$ in $K$-class classification is computationally expensive, as it requires evaluation over $2^K$ subsets—akin to computing the generalised Hartley measure. To address this bottleneck, Proposition 21 introduces a practical linear-time upper bound. The key idea is to approximate any $\underline{P}$ with the closest and uniformly more conservative $\epsilon$ contamination model $\underline{P}_\epsilon$, i.e. $\underline{P}_\epsilon(A) \leq \underline{P}(A)$ for all events $A$, ensuring no additional certainty is introduced for this approximation. Leveraging Axiom A3 in Theorem 20 and Proposition 19, we derive a computationally efficient upper bound for $\mathrm{MMI}_{\mathcal{F}_{TV}}(\underline{P})$, relaxing computation to $O(K)$.

**Proposition 21.** *Let $\mathcal{X}$ be finite. For any $\underline{P} \in \underline{\mathcal{P}}(\mathcal{X})$, $\mathrm{MMI}_{\mathcal{F}_{TV}}(\underline{P}) \leq 1 - \sum_{x \in \mathcal{X}} \underline{P}(\{x\})$.*

## 5 Related work

**Distances and Divergences for IPs.** Several metrics and divergences have been proposed for credal sets [102–106], and were recently categorised by Chau et al. [34] into inclusion, equality, and intersection measures. Given its connection with lower probabilities, our IIPM belongs to the equality class. For fuzzy sets and fuzzy measures, see Couso et al. [107], Montes et al. [108]. Specifically for lower probabilities, Hable [109] studied minimum distance estimation between a precise parametric model and a lower probability model, using a special case of $\mathrm{IIPM}_{\mathcal{F}}$ with bounded continuous functions over a discretised domain—though without theoretical analysis. In the context of belief functions, divergences such as those in [110–112] operate on the möbius transform of belief functions, while the Minkowski measure [113] compares belief functions via summation of all assessments. More recently, Xiao et al. [114] introduced a generalised $f$-divergence for belief functions. Closest in spirit to our work is Catalano and Lavenant [115], who proposed an integral-based metric for higher-order probabilities.

**Uncertainty Quantification for IPs.** To quantify epistemic—and occasionally aleatoric—uncertainty in credal-set-based models, several measures have been proposed, including maximal entropy [116], entropy difference [96], the generalised Hartley measure [94], and credal set volume [117]. In hierarchical or second-order models such as Bayesian and evidential frameworks, entropy-based measures—including entropy, conditional entropy, and mutual information—have been widely used to capture total, epistemic, and aleatoric uncertainty [118, 92]; see Wimmer et al. [119], Smith et al. [120] for critiques. Alternatively, Sale et al. [121, 122] advocate variance and probability distances as general-purpose EU measures for second-order models. A recent review is provided by Hoarau et al. [123].

Overall, to our knowledge, our work is the first to introduce a Choquet-integral-based metric framework for comparing capacities—a foundational class for many uncertainty models—marking a novel use of integral-based distances for quantifying credal uncertainty.

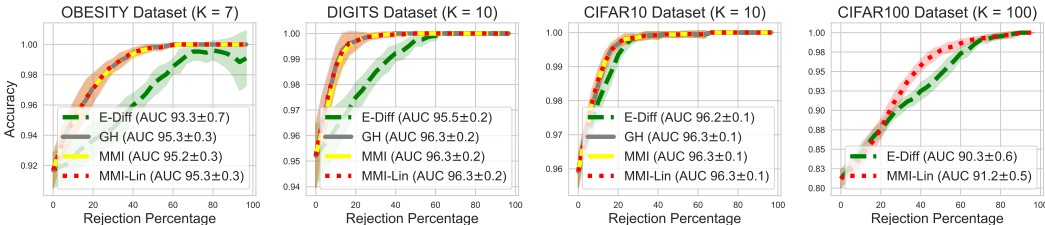

**Figure 1:** Accuracy-Rejection (AR) curves on four classification tasks. The area under the curve (AUC) is reported for numerical comparison. We consistently outperform entropy difference (`E-Diff`) and match the performance of Generalised Hartley (`GH`). On large-scale problems, our efficient upper bound (`MMI-Lin`) remains tractable and continues to outperform `E-Diff`.

## 6 Empirical validation of MMI with selective prediction experiments

This section demonstrates the practicality of the proposed MMI measure. Additional ablation studies and detailed experiment descriptions are provided in Appendix C. The code to reproduce our experiments is here [124]. We evaluate MMI by its ability to capture informative EU in $K$-class classification. As ground-truth uncertainty is unavailable, the informativeness of epistemic UQ is typically assessed by its utility in improving prediction or decision-making [125, 126]. Following Shaker and Hüllermeier [80], Hüllermeier et al. [96] while extending their binary classification experiments to the multiclass case, we evaluate our performance by plotting the *accuracy-rejection* (AR) curves in selective classification problems. An AR curve plots accuracy against rejection rate: a model that abstains on $p\%$ of inputs and predicts on the most certain $(1 - p)\%$ should show increasing accuracy with $p$. An informative EU measure yields a monotonic AR curve, approaching the top-left corner—analogous to ROC curves in standard classification.

**Setup.** As we are working with discrete label spaces, we consider $\mathcal{F}_{TV}$ as our function class for MMI. Its linear-time upper bound is denoted as `MMI-Lin`. We compare them against two popular epistemic UQ measures: the entropy difference (`E-Diff`) [100] and the generalised Hartley measure (`GH`) [127]. To construct the credal set and corresponding lower probabilities, we utilise $m = 10$ probabilistic predictors trained with different hyperparameter configurations—random forests for tabular data and neural networks for image data—and make predictions using the centroid of the credal set, equivalent to averaging over all predictors. Specifically, for an instance $x$ and predictors $\{\hat{P}_j(Y \mid X = x)\}_{j=1}^m$, the lower probability is constructed as $\underline{P}(A) = \min_{j \in \{1,\ldots,m\}} \hat{P}_j(Y \in A \mid X = x)$ for every possible subset of classes, and predictions are made using $\frac{1}{m}\sum_{j=1}^m \hat{P}_j(Y \mid X = x)$. While more advanced aggregation methods exist [14, 48], our aim is to assess UQ measures, not optimise prediction. The lower probability here encodes the EU information we wish to recover using MMI.The lower probability captures the epistemic uncertainty that `MMI` seeks to quantify. We evaluate on two UCI datasets [128, 129] and CIFAR-10/100 [130]. We perform train-test splits on the datasets and produce AR curves on the latter. All experiments are repeated 10 times, and the mean and standard deviation of the accuracies are plotted. The full experimental set-up is discussed in Appendix C.

**Results.** Figure 1 presents AR curves for our classification problems with 7, 10, and 100 classes. The plots share the same message: `MMI` and `MMI-Lin` consistently outperform `E-Diff` and perform almost identical to `GH`, which is often considered the most informative measure for epistemic UQ [123]. This equivalence in performance may be because both measures satisfy all 6 desirable axioms for epistemic UQ measures. Notably, Hoarau et al. [123] showed that `GH` and the credal epistemic measure (i.e., `MMI` when $K = 2$) exhibit perfect correlation, suggesting they encode identical information. Our results extend this insight, indicating that this equivalence may hold for arbitrary $K$. For large $K$, where exact computation of `MMI` and `GH` becomes intractable ($O(2^K)$), the upper bound `MMI-Lin` remains efficient ($O(K)$) and continues to outperform `E-Diff`, offering a practical and reliable alternative to `GH`. These results clearly demonstrate that our proposed IIPM framework and its application as an epistemic UQ measure are not only theoretically justified, but also practical.

# 7 Discussion, limitations, and future directions

We introduced Integral Imprecise Probability Metrics (IIPMs) as a generalisation of classical Integral Probability Metrics (IPMs) to the setting of imprecise probabilities. This framework enables the comparison of IP models through Choquet integration. We provided initial theoretical analyses and illustrated their practicality through epistemic UQ. Notably, the framework gives rise to a new class of epistemic UQ measures—termed Maximum Mean Imprecisions—which outperforms several popular alternatives in empirical evaluations on selective classification problems.

While this work opens new avenues for IPML research, a key limitation is the lack of a sampling-based estimation method for IIPMs, unlike in classical IPMs. This arises from the absence of a canonical sampling procedure for imprecise models, which capture subjective beliefs rather than event frequencies. Nonetheless, advancing this direction could, for example, enable the integration of EU into generative models based on IPMs [131, 72]. On the theoretical side, it would be valuable to connect and extend other divergence classes, such as $f$-divergences and Bregman divergences, to the context of capacities and more general IPs. Another open question is whether weaker conditions on the function class $\mathcal{F}$ or space $\mathcal{X}$ can ensure that $\mathrm{IIPM}_{\mathcal{F}}$ metrises capacities. From a computational perspective, developing efficient nonparametric estimation methods for IIPMs is a key priority. A further application-oriented direction is to quantify credal epistemic uncertainty in regression problems using the IIPM framework, particularly in light of recent proposals for uncertainty quantification axioms in regression [132].

**Acknowledgement.** The authors would like to thank Anurag Singh, Shireen Kudukkil Manchingal, and Fabio Cuzzolin for insightful discussions. Special thanks to Masha Naslidnyk for her detailed proofreading and thoughtful feedback. The authors also gratefully acknowledge the University of Manchester and CISPA Helmholtz Center for Information Security for supporting Siu Lun's research visit to Manchester, without which this project would not have been possible.

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

# Appendix of *Integral Imprecise Probability Metrics*

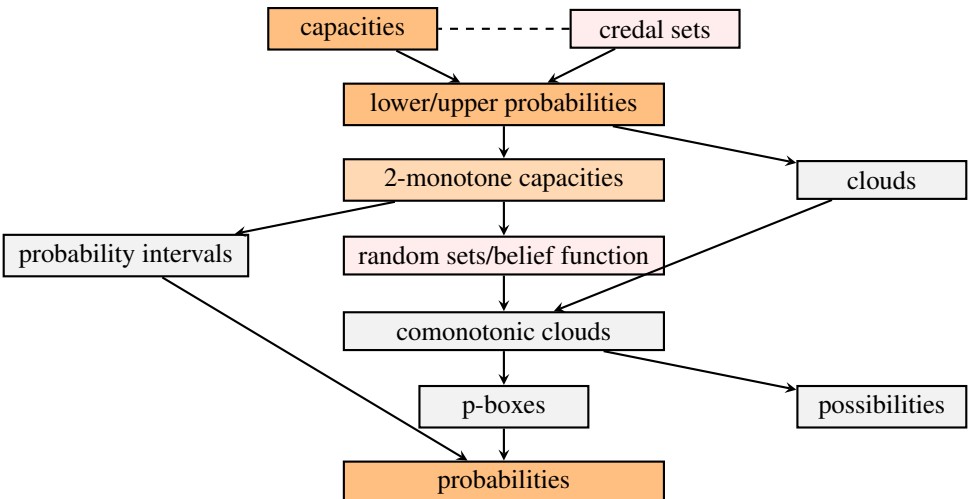

**Figure 2:** A skeleton demonstrating the connection between various uncertainty calculi. "A → B" means A generalises B, meaning that B is a specific instance of A. The figure is adopted from Destercke et al. [133] and Hüllermeier and Waegeman [4]. Most of these frameworks generalise classical probability theory. In the main text, we have discussed capacities, lower and upper probabilities, and standard probabilities in detail, with a brief mention of 2-monotone capacities. Additional discussion of credal sets and belief functions is provided in Appendix A. We do not elaborate on the methods shown in grey; for those, we refer readers to existing surveys and review articles.

# A   Some more introduction to Imprecise Probabilistic Machine Learning (IPML)

In the main text, we demonstrate the theoretical appeal and practical utility of IIPM primarily through capacities and lower probabilities. While these models are already quite general, we complement this focus by discussing two other mainstream approaches in IPML—credal sets and belief functions—that, although mathematically related, are conceptually motivated in distinct ways. As illustrated in the hierarchy in Figure 2, these models are closely connected to the core ideas of the paper and further support the relevance and broad applicability of the proposed IIPM and MMI framework.

## A.1   What is Imprecise Probabilistic Machine Learning actually doing?

At its core, probabilistic machine learning seeks to construct mathematical models that, through data-driven learning procedures, capture underlying physical or real-world phenomena. For instance, in generative modelling, the objective is to learn the marginal probability distribution that governs the data-generating process. In predictive tasks, instead, the goal is to estimate the conditional distribution of a target variable $Y$ given an input $x$—or its expectation in the case of regression. We often refer to these kinds of natural variation and randomness as aleatoric uncertainty.

Imprecise probabilistic machine learning (IPML) extends this foundation by moving beyond the exclusive use of precise probability models. While classical probability excels at modelling aleatoric uncertainty, IPML incorporates imprecise probability models to account not only for inherent randomness but also for epistemic uncertainty—allowing ambiguity, partial knowledge, and doubt to be explicitly represented within the model. For readers interested in deeper treatments on IP, we recommend Cuzzolin [9] for a comprehensive introduction, Hüllermeier and Waegeman [4, Appendix A], for a concise overview, and Caprio et al. [47, Appendix A], for a discussion on why we should care about imprecision.

In the following, we provide an overview of two other mainstream modelling approaches in IPML: credal set and belief function approaches. We outline the motivations behind these methods and provide some examples of how they are integrated into ML to improve uncertainty quantification and predictive performance.

## A.2 Credal sets and their use in IPML

Credal sets sit at the top of the hierarchy shown in Figure 2, making them one of the most general constructs in imprecise probability. Many other models can be seen as special cases of credal sets endowed with additional structure. Consequently, there are numerous ways to construct a credal set, depending on the modelling assumptions and information available.

Credal sets are generally understood as some convex set of probability measures $\mathcal{C} \subset \mathcal{P}(\mathcal{X})$. Convexity can be justified in different ways. In Quasi-Bayesian decision theory [134], it can be shown that rationality axioms proposed for **partial** binary preference naturally leads to a convex set of probability measures, akin to how Savage [135] showed rational binary preference leads to the existence of a unique single probability distribution (paired along with an utility function). Alternatively, in formal epistemology, Williamson [136] put forward the notion of *Chance Calibration*, which is also closely related to Lewis [7]'s *Principal Principle*, which puts into the words of a statistician means, when, according to the current observations, the actual distribution of interest encoding physical phenomena lies within some set $\{P_1, \ldots, P_m\}$ of distributions, but the modeller is indifferent as to which distribution, then rationally, the modeller's belief, also represented as a distribution, should lie within $\mathrm{ConvexHull}(\{P_1, \ldots, P_m\})$. This means any distribution in the convex hull is considered a rational belief, thus, the set itself captures the set of rational beliefs, encompassing our imprecision.

In IPML, credal sets are often used to represent either

1. Dataset/Distribution-level uncertainty, or

2. Predictive uncertainty.

**Case 1.** In the first case, examples include distributionally robust optimisation[137], where a learning algorithm is optimised against the worst-case empirical risk over a credal set—a set of distributions within an $\epsilon$-distance from the observed empirical distribution. In the out-of-domain generalisation literature, given observations from multiple source distributions $\mathbb{P}_1, \ldots, \mathbb{P}_m$, it is commonly assumed that the test-time distribution lies within their convex hull [138, 139], which effectively forms a credal set. Singh et al. [14] made this connection explicit and proposed an algorithm that allows the test-time ambiguity to be resolved without additional training. Caprio et al. [49] developed a learning-theoretic framework for supervised learning under credal sets, while Chau et al. [15] introduced a hypothesis testing procedure for statistically comparing credal sets.

**Case 2.** In the second case, credal sets are used to model epistemic uncertainty in prediction. In credal Bayesian deep learning (CBDL) [47], finitely generated credal sets over priors and likelihoods are combined, by considering all combinatorial applications of Bayes' rule, to yield a posterior credal set. Wang et al. [83] introduced credal-set interval neural networks, which predict credal sets from probability intervals derived from deterministic interval neural network outputs. Similarly, for classification tasks, Wang et al. [140] proposed defining a predictive credal set as the collection of probability vectors within the simplex that satisfy lower and upper bounds on class probabilities, derived from a set of probabilistic predictors.

## A.3 Belief function/random set and their use in IPML

Moving down the hierarchy—and skipping lower probabilities and 2-monotone capacities, which are already discussed in the main paper—we briefly describe the roles of random set theory and belief functions.

We focus on finite instance spaces $\mathcal{X}$, where random set theory and belief functions coincide. Our exposition follows the terminology of Shafer's seminal work on The Theory of Evidence [23] and the overview in Cuzzolin [141, Chapter 2.2]. The core philosophy behind belief function theory is its emphasis on representing degrees of support—capturing epistemic uncertainty—rather than specifying how these values are generated, which relates more to aleatoric uncertainty. Perhaps more importantly are the tools developed to combine multiple evidence in a coherent manner.

So, how are belief functions defined? We first introduce a fundamental concept, called basic probability assignments. Let $\mathcal{X}$ be finite.

**Definition 22.** *A basic probability assignment over $\mathcal{X}$ is a set function $m : 2^{\mathcal{X}} \to [0,1]$ such that*

$$m(\emptyset) = 0, \quad \sum_{A \subseteq \mathcal{X}} m(A) = 1.$$

The subsets that have non-zero mass are known as the focal elements within $2^{\mathcal{X}}$. Basic probability assignment happens in practice when, e.g., sensors have limited precision and can only give results of the type "A or B" [142].

Given a mass function $m$, which intuitively represents the degree of belief that the true outcome lies exactly within the subset $A \in 2^{\mathcal{X}}$, $m(A)$ quantifies the support assigned to the set $A$ and no more specific subset. From here, we can derive the belief function.

**Definition 23.** *The belief function associated with a basic probability assignment $m : 2^{\mathcal{X}} \to [0,1]$ is the set function $\mathrm{Bel} : 2^{\mathcal{X}} \to [0,1]$ defined as,*

$$\mathrm{Bel}(A) = \sum_{B \subseteq A} m(B).$$

Its conjugate, known as the plausibility function Pl, measures the amount of evidence *not against an event $A$* by measuring the following,

$$\mathrm{Pl}(A) = 1 - \mathrm{Bel}(A^c).$$

The theory of evidence is rich and conceptually deep, and a full treatment lies beyond the scope of this appendix. However, the context provided should suffice to understand a recent integration of belief functions into machine learning. Specifically, Manchingal et al. [143] introduced a new class of neural networks, called Random-Set Neural Networks (RS-NNs), for $K$-class classification. Instead of producing a probability vector with $K$ outputs—as in standard classifiers—RS-NNs are designed to output a basic probability assignment over the $K$ classes, requiring $2^K$ output nodes to represent belief mass on all subsets of classes. Since this is computationally infeasible for large $K$, a preprocessing step is introduced to select a subset of focal elements, including the original singleton classes and additional relevant subsets. Given the resulting mass function, belief and plausibility functions can then be derived for prediction and uncertainty quantification. This principle is further extended to convolutional neural networks in Manchingal et al. [81]. As perhaps the first of its kind, this random set-based learning paradigm has also been recently adapted to large language models in Mubashar et al. [144], offering an explicit mechanism for modelling epistemic uncertainty in LLMs.

## B   Proofs and derivations

This section presents the proofs and derivation in the main text.

### B.1   Proof of Lemma 5

**Lemma 5.** *For lower probability $\underline{P}$ associated to credal set $\mathcal{C}$, we have $\oint f d\underline{P} \leq \inf_{P \in \mathcal{C}} \int f dP$ for any $f \in C_b(\mathcal{X})$. When $\underline{P}$ is 2-monotonic, the inequality becomes an equality.*

*Proof.* Let $\underline{P}$ be the lower probability associated to the credal set $\mathcal{C}$. For $f \in C_b(\mathcal{X})$, we can write the Choquet integral as,

$$\oint f d\underline{P} = \underline{f} + \int_{\underline{f}}^{\overline{f}} \underline{P}(\{f \geq t\}) dt$$

$$= \underline{f} + \int_{\underline{f}}^{\overline{f}} \inf_{P \in \mathcal{C}} P(\{f \geq t\}) dt$$

$$\leq \underline{f} + \inf_{P \in \mathcal{C}} \int_{\underline{f}}^{\overline{f}} P(\{f \geq t\}) dt$$

$$= \inf_{P \in \mathcal{C}} \int f dP.$$

For 2-monotone $\underline{P}$, the results follow from Delbaen [145, Lemma 2]. $\qquad\square$

## B.2 Proof of Theorem 6

**Theorem 6.** *Let $(\mathcal{X}, d)$ be a metric space. For any capacities $\nu, \mu \in \mathcal{V}(\mathcal{X})$, we have $\oint f d\nu = \oint f d\mu$ for all $f \in C_b(\mathcal{X})$, if and only if $\nu = \mu$.*

*Proof.* ($\implies$) Let $U$ be any open set in $\mathcal{X}$ and $F$ the complement. Consider the distance $d(x, F) = \min_{y \in F} d(x, y)$. For $n = 1, 2, \ldots$, let $f_n(x) = \min(1, nd(x, F))$. Then, $\sup_{x \in S} |f_n - f| \to 0$, where $f = \mathbf{1}_U$ is the indicating function. Now, as Choquet integration is continuous with respect to the topology of uniform convergence [19, Proposition C.5(ix)], we have

$$\lim_{n \to \infty} \oint f_n d\nu = \oint \mathbf{1}_U d\nu = \oint \mathbf{1}_U d\mu.$$

This implies $\nu(U) = \mu(U)$. Now, since $(\mathcal{X}, d)$ is a metric space, for any $A \in \Sigma_{\mathcal{X}}$, we can find an increasing sequence of open subsets $A_1 \subseteq A_2, \ldots$ such that $A_n \uparrow A$. Since $\nu, \mu$ are continuous from below, we have $\lim_{n \to \infty} \nu(A_n) = \nu(A)$, but since for any open set $\nu(A_n) = \mu(A_n)$, we conclude $\nu(A) = \mu(A)$, thus $\nu = \mu$.

($\impliedby$) If $\nu = \mu$, then it is trivial to see $\oint f d\nu = \oint f \mu$ for any $f \in C_b(S)$. $\qquad\square$

## B.3 Proof of Corollary 8

**Corollary 8.** *For any $P, Q \in \mathcal{P}(\mathcal{X})$ and $\mathcal{F} \subseteq C_b(\mathcal{X})$, $\mathrm{IIPM}_{\mathcal{F}}(P, Q) = \mathrm{IPM}_{\mathcal{F}}(P, Q)$.*

*Proof.* For any $P, Q \in \mathcal{P}(\mathcal{X})$ and $\mathcal{F} \subseteq \mathcal{C}_b(\mathcal{X})$, we have

$$\mathrm{IIPM}_{\mathcal{F}}(P, Q) = \sup_{f \in \mathcal{F}} \left\{ \left| \oint f dP - \oint f dQ \right| \right\}$$

$$= \sup_{f \in \mathcal{F}} \left\{ \left| \int f dP - \int f dQ \right| \right\}$$

$$= \mathrm{IPM}_{\mathcal{F}}(P, Q),$$

since the Choquet integral for additive probability is the Lebesgue integral. $\qquad\square$

## B.4 Proof of Proposition 9

**Proposition 9.** *For any $\mathcal{F} \subseteq C_b(\mathcal{X})$, $\mathrm{IIPM}_{\mathcal{F}}$ is a pseudometric on $\mathcal{V}(\mathcal{X})$; it is **non-negative**, **symmetric**, and satisfies the **triangle inequality**.*

*Proof.* To prove it is a pseudometric, we need non-negativity, symmetry, and to show triangle inequality.

- **Non-negative:** It is obvious that $\mathrm{IIPM}_{\mathcal{F}}(\nu_1, \nu_2) \geq 0$ for any pair of $\nu_1, \nu_2 \in V(\mathcal{X})$

- **Symmetric:** Symmetry is also apparent.

- **Triangle inequality:** Pick $\nu_1, \nu_2, \nu_3$ from $V(\mathcal{X})$, then

$$\mathrm{IIPM}_{\mathcal{F}}(\nu_1, \nu_2) = \sup_{f \in \mathcal{F}} \left| \oint f d\nu_1 - \oint f d\nu_2 \right|$$

$$= \sup_{f \in \mathcal{F}} \left| \oint f d\nu_1 - \oint f d\nu_3 + \oint f d\nu_3 - \oint f d\nu_2 \right|$$

$$\leq \sup_{f \in \mathcal{F}} \left| \oint f d\nu_1 - \oint f d\nu_3 \right| + \sup_{f \in \mathcal{F}} \left| \oint f d\nu_3 - \oint f d\nu_2 \right|$$

$$= \mathrm{IIPM}_{\mathcal{F}}(\nu_1, \nu_3) + \mathrm{IIPM}_{\mathcal{F}}(\nu_3, \nu_2).$$

This concludes the proof. We note that in some literature, pseudometric also requires $\mathrm{IIPM}_{\mathcal{F}}(\nu, \nu) = 0$, and this also trivially holds in our case. $\qquad\square$

### B.5   Proof of Theorem 10

**Theorem 10.** *Let $\mathcal{F} \subseteq C_b(\mathcal{X})$ be dense in $C_b(\mathcal{X})$ with respect to the $\| \cdot \|_\infty$ norm. Then, $\mathrm{IIPM}_\mathcal{F}$ metrises the Choquet weak convergence of $\mathcal{V}(\mathcal{X})$.*

*Proof.* To show that $\mathrm{IIPM}_\mathcal{F}$ metrises the Choquet weak convergence of $\mathcal{V}(\mathcal{X})$, we need to show for $\nu_n, \nu \in \mathcal{V}(\mathcal{X})$, whenever $\mathrm{IIPM}_\mathcal{F}(\nu_n, \nu) \to 0$ then $\nu_n$ converges to $\nu$ in the Choquet weak sense.

Now pick $f \in C_b(\mathcal{X})$, since by assumption $\mathcal{F}$ is dense in $C_b(\mathcal{X})$, there exists $g \in \mathcal{F}$ satisfying $\|f - g\|_\infty < \epsilon$. By assumption, $\mathrm{IIPM}_\mathcal{F}(\nu_n, \nu) \to 0$ means $|\oint g d\nu_n - \oint g d\nu| \to 0$ since $g \in \mathcal{F}$. Thus, we have the bound

$$\left| \oint f d\nu_n - \oint f d\nu \right| \leq \left| \oint f d\nu_n - \oint g d\nu_n \right| + \left| \oint g d\nu_n - \oint g d\nu \right| + \left| \oint g d\nu - \oint f d\nu \right|$$

$$\leq 2\epsilon$$

for all $f \in C_b(\mathcal{X})$ and $\epsilon > 0$, which implies $\nu_n$ converges to $\nu$ in the Choquet weak sense.

Proving the other direction is a direct application of the result from Theorem 6 since $\mathcal{F} \subseteq C_b(\mathcal{X})$. $\square$

### B.6   Proof of Proposition 12

**Proposition 12.** *The Lower Dudley metric metrises Choquet weak convergence on $\underline{\mathcal{P}}(\mathcal{X})$.*

*Proof.* The core idea is to show that $\mathcal{F}_d$ is dense in $C_b(\mathcal{X})$ and then to apply Theorem 10.

To show denseness, pick any $f \in C_b(\mathcal{X})$, consider the sequence

$$f_n(x) = \inf_{y \in \mathcal{X}} \{ f(y) + nd(x, y) \}.$$

Pick $\alpha > 0$ such that $|f(x)| \leq \alpha$ for all $x \in \mathcal{X}$, thus $|f(x) - f(y)| \leq |f(x)| + |f(y)| \leq \alpha$ for any $x, y \in \mathcal{X}$. It's clear that $-\alpha \leq f_n \leq f$ is bounded and $n-$Lipschitz continuous. Now we prove $f_n \to f$ uniformly. Fix $\epsilon > 0$, there is $\delta > 0$ such that $d(x, y) < \delta$ implies $|f(x) - f(y)| < \epsilon$ since $f$ is continuous. Then for all $x \in \mathcal{X}$, we have

$$0 \leq f(x) - f_n(x) = f(x) - \inf_{y \in \mathcal{X}} \{ f(y) + nd(x, y) \}$$

$$= \sup_{y \in \mathcal{X}} \{ f(x) - f(y) - nd(x, y) \}$$

$$= \sup_{y \in \mathcal{X}, d(x,y) \leq 2\alpha/n} \{ f(x) - f(y) - nd(x, y) \}.$$

If $n$ is such that $\frac{2\alpha}{n} < \delta$, then

$$f(x) - f_n(x) \leq \sup\{ \epsilon - nd(x, y) \mid y \in \mathcal{X} \text{ s.t. } d(x, y) \leq 2\alpha/n \} \leq \epsilon$$

which follows that $\|f - f_n\| \leq \epsilon$, meaning $\mathcal{F}_d$ is dense in $C_b(\mathcal{X})$ in the uniform norm. $\square$

### B.7   Proof of Remark 14

**Remark 14.** *Let $\mathcal{X}$ be finite. For any $P, Q \in \mathcal{P}(\mathcal{X})$, we have $\mathrm{IPM}_{\mathcal{F}_{TV}}(P, Q) = \sup_{A \in \Sigma_\mathcal{X}} |P(A) - Q(A)| = \frac{1}{2} \sum_{x \in \mathcal{X}} (P(\{x\}) - Q(\{x\}))$. In contrast, in the imprecise case, there exists $\underline{P}, \underline{Q} \in \underline{\mathcal{P}}(\mathcal{X})$ such that $\mathrm{IIPM}_{\mathcal{F}_{TV}}(\underline{P}, \underline{Q}) := \sup_{A \in \Sigma_\mathcal{X}} |\underline{P}(A) - \underline{Q}(A)| \neq \frac{1}{2} \sum_{x \in \mathcal{X}} (\underline{P}(\{x\}) - \underline{Q}(\{x\}))$.*

*Proof.* We provide an example based on Montes et al. [146, Example 4] for completeness. Consider $\mathcal{X} = \{x_1, x_2, x_3\}$ and lower probabilities $\underline{P}_1, \underline{P}_2$ given by

| | $\emptyset$ | $\{x_1\}$ | $\{x_2\}$ | $\{x_3\}$ | $\{x_1, x_2\}$ | $\{x_1, x_3\}$ | $\{x_2, x_3\}$ | $\mathcal{X}$ |
|---|---|---|---|---|---|---|---|---|
| $\underline{P}_1$ | 0 | 0 | 0 | 0 | $1/2$ | $1/2$ | $1/2$ | 1 |
| $\underline{P}_2$ | 0 | 0 | 0 | 0 | $1/3$ | $1/3$ | $1/3$ | 1 |

Then, we know $d_1 = \sup_{A \subseteq \mathcal{X}} |\underline{P}_1(A) - \underline{P}_2(A)| = 1/6$, whereas $d_2 = \sum_{x \in \mathcal{X}} |\underline{P}_1(\{x\}) - \underline{P}_2(\{x\})| = 0$, therefore $d_1 \neq d_2$. $\square$

## B.8  Proof of Lemma 15

**Lemma 15.** *Let $\mathcal{P}_{\epsilon,P}$ be an $\epsilon$-contaminated model defined in Equation* (2). *Then, the associated lower probability $\underline{P}_\epsilon$ is given by*

$$\underline{P}_\epsilon(A) = \inf_{\tilde{P}\in\mathcal{P}_{\epsilon,P}} \tilde{P}(A) = \begin{cases} (1-\epsilon)P(A) & \text{for all } A \in \Sigma_{\mathcal{X}}\backslash\{\mathcal{X}\} \\ 1, & \text{for } A = \mathcal{X} \end{cases}$$

*Proof.* This lemma is proved in Walley [8, Section 2.9.2]. $\qquad\square$

## B.9  Proof of Theorem 16

**Theorem 16.** *Let $\mathcal{F}_W := \{f \in C_b(\mathcal{X}) : \|f\|_L \leq 1\}$ where $\|f\|_L := \sup_{x,y\in\mathcal{X}}\{|f(x)-f(y)|/c(x,y)\}$, and $c$ the transportation cost in a restricted lower probability Kantorovich (RLPK) problem [56, Definition 10]. Let $\underline{P}_\epsilon$, $\underline{Q}_\epsilon$ be lower probabilities of the $\epsilon$-contaminated models $\mathcal{P}_{\epsilon,P}$ and $\mathcal{P}_{\epsilon,Q}$. Then, $\mathrm{IIPM}_{\mathcal{F}_W}(P,Q)$ coincides with the objective of the RLPK problem, and thus coincides with the classical Kantorovich's optimal transport problem involving $P$ and $Q$.*

*Proof.* Our goal is to show that $\mathrm{IIPM}_{\mathcal{F}_W}(\underline{P}_\epsilon, \underline{Q}_\epsilon)$ coincides with the objective of the restricted lower probability Kantorovich problem. Note that we have,

$$
\begin{aligned}
\mathrm{IIPM}_{\mathcal{F}_W}(\underline{P}_\epsilon, \underline{Q}_\epsilon) &= \sup_{f\in\mathcal{F}_W} \left| \oint f d\underline{P}_\epsilon - \oint f d\underline{Q}_\epsilon \right| \\
&= \sup_{f\in\mathcal{F}_W} \left| \int_{\underline{f}}^{\overline{f}} [\underline{P}_\epsilon(\{f \geq t\}) - \underline{Q}_\epsilon(\{f \geq t\})]dt \right| \\
&\overset{(\heartsuit)}{=} \sup_{f\in\mathcal{F}_W} \left| \int_{\underline{f}}^{\overline{f}} [\underline{P}_\epsilon(\{f > t\}) - \underline{Q}_\epsilon(\{f > t\})]dt \right| \\
&= \sup_{f\in\mathcal{F}_W} \left| \int_{\underline{f}}^{\overline{f}} [(1-\epsilon)P(\{f > t\}) - (1-\epsilon)Q(\{f > t\})]dt \right| \\
&\overset{(\clubsuit)}{\leq} (1-\epsilon) \sup_{f\in\mathcal{F}_W} \left| \int f dP - \int f dQ \right| \\
&\overset{(\spadesuit)}{=} (1-\epsilon) \inf_{\pi\in\Gamma(P,Q)} \int c(x,y)\pi(dx,dy)
\end{aligned}
$$

where $\Gamma(P,Q)$ is the set of joint probability with marginals being $P$ and $Q$. We replaced the inequality with strict inequality in $\heartsuit$ as in Troffaes and De Cooman [19, Proposition C.3.ii]. In $\clubsuit$ we used the fact that Choquet integration returns the standard Lebesgue integral when the capacity is a probability measure, and in $\spadesuit$ we used Kantorovich-Rubinstein theorem [147, Lecture 3], which established the duality between the Kantorovich problem and an IPM formulation using function class $\mathcal{F}_W$. This recovered the result from Caprio [56] through the use of our IIPM framework. $\qquad\square$

## B.10  Proof of Proposition 18

**Proposition 18.** *The definition of MMI is equivalent to*

$$\mathrm{MMI}_{\mathcal{F}}(\underline{P}) = \sup_{f\in\mathcal{F}} \int_{\underline{f}}^{\overline{f}} 1 - \left( \underline{P}(\{f < t\}) + \underline{P}(\{f \geq t\}) \right) dt \tag{3}$$

*Proof.* To show the result, it follows from the definition that

$$\mathrm{MMI}_{\mathcal{F}}(\underline{P}) = \sup_{f \in \mathcal{F}} \left\{ \oint f d\overline{P} - \oint f d\underline{P} \right\}$$

$$= \sup_{f \in \mathcal{F}} \int_{\underline{f}}^{\overline{f}} [\overline{P}(\{f \geq t\}) - \underline{P}(\{f \geq t\})] dt$$

$$= \sup_{f \in \mathcal{F}} \int_{\underline{f}}^{\overline{f}} [1 - (\underline{P}(\{f < t\}) + \underline{P}(\{f \geq t\}))] dt,$$

which completes the proof. $\qquad\square$

### B.11  Proof of Proposition 19

**Proposition 19** (MMI on $\epsilon$-contamination set.)**.** *Let $\underline{P}_\epsilon$ be the lower probability associated with $\mathcal{P}_{\epsilon,P}$ and $\mathcal{F} \subseteq C_b(\mathcal{X})$. Then $\mathrm{MMI}_{\mathcal{F}}(\underline{P}_\epsilon) = \epsilon \left( \sup_{f \in \mathcal{F}} \sup_{x,y \in \mathcal{X}} |f(x) - f(y)| \right)$. For the LTV distance with $\mathcal{F}_{TV} := \{\mathbf{1}_A : A \in \Sigma_{\mathcal{X}}\}$, we have $\mathrm{MMI}_{\mathcal{F}_{TV}}(\underline{P}_\epsilon) = \sup_{A \in \Sigma_{\mathcal{X}}} \{\overline{P}_\epsilon(A) - \underline{P}_\epsilon(A)\} = \epsilon$.*

*Proof.* First, it can be shown readily the upper probability model for an $\epsilon$ contamination set:

$$\overline{P}(A) = \begin{cases} (1 - \epsilon)P(A) + \epsilon & \text{for } A \in \Sigma_{\mathcal{X}} \backslash \emptyset \\ 0 & \text{for } A = \emptyset \end{cases}.$$

Next, we have

$$\mathrm{MMI}_{\mathcal{F}}(\underline{P}_\epsilon) = \sup_{f \in \mathcal{F}} \left\{ \oint f d\overline{P} - \oint f d\underline{P} \right\}$$

$$= \sup_{f \in \mathcal{F}} \int_{\underline{f}}^{\overline{f}} [\overline{P}(\{f \geq t\}) - \underline{P}(\{f \geq t\})] dt$$

$$= \sup_{f \in \mathcal{F}} \int_{\underline{f}}^{\overline{f}} [\epsilon + (1 - \epsilon)P(\{f > t\}) - (1 - \epsilon)P(\{f > t\})] dt$$

$$= \sup_{f \in \mathcal{F}} \int_{\underline{f}}^{\overline{f}} \epsilon \, dt$$

$$= \epsilon \left( \sup_{f \in \mathcal{F}} [\overline{f} - \underline{f}] \right) = \epsilon \left( \sup_{f \in \mathcal{F}} \sup_{x,y \in \mathcal{X}} |f(x) - f(y)| \right),$$

which shows the result. For $\mathcal{F}_{TV}$, it is straightforward to see that the maximum value the terms in the bracket of the last equation can attain is 1. Therefore,

$$\mathrm{MMI}_{\mathcal{F}_{TV}}(\underline{P}_\epsilon) = \epsilon.$$

This completes the proof. $\qquad\square$

### B.12  Proof for Theorem 20

To prove our uncertainty measure satisfies the axioms, we need the following useful lemma.

**Lemma 24** (Marginalisation preserves 2-monotonicty.)**.** *If $\underline{P}$ is a 2-monotone capacity defined on the joint measurable space $(\mathcal{X} \times \mathcal{Y}, \Sigma_{\mathcal{X}} \times \Sigma_{\mathcal{Y}})$, and let $\underline{P}_1(\cdot) = \underline{P}(\cdot, \mathcal{Y})$ be the marginal capacity on $(\mathcal{X}, \Sigma_{\mathcal{X}})$ and $\underline{P}_2(\cdot) = \underline{P}(\mathcal{X}, \cdot)$ be the marginal capacity on $(\mathcal{Y}, \Sigma_{\mathcal{Y}})$, then $\underline{P}_1$ and $\underline{P}_2$ are also 2-monotone. Marginalisation also preserves 2-alternating.*

*Proof.* Recall the definition of 2-monotonicty, for any $A, B \in \Sigma_{\mathcal{X}} \times \Sigma_{\mathcal{Y}}$, we have

$$\underline{P}(A \cup B) + \underline{P}(A \cap B) \geq \underline{P}(A) + \underline{P}(B).$$

Now pick $A_1, B_1 \in \Sigma_{\mathcal{X}}$, then consider,

$$
\begin{aligned}
\underline{P}_1(A_1 \cup B_1) + \underline{P}_1(A_1 \cap B_1) &= \underline{P}((A_1 \cup B_1) \times \mathcal{Y}) + \underline{P}((A_1 \cap B_1) \times \mathcal{Y}) \\
&= \underline{P}((A_1 \times \mathcal{Y}) \cup (B_1 \times \mathcal{Y})) + \underline{P}((A_1 \times \mathcal{Y}) \cap (B_1 \times \mathcal{Y})) \\
&\geq \underline{P}(A_1 \times \mathcal{Y}) + \underline{P}(B_1 \times \mathcal{Y}) \\
&= \underline{P}_1(A_1) + \underline{P}_1(B_1).
\end{aligned}
$$

This shows that 2-monotonicity holds for $\underline{P}_1$ and by symmetry, it also holds for $\underline{P}_2$. Therefore, marginalisation preserves 2-monotonicity. The steps to show marginalisation preserves 2-alternating are analogous. $\qquad \square$

**Theorem 20.** *For any $\mathcal{F} \subseteq C_b(\mathcal{X})$, $\mathrm{MMI}_{\mathcal{F}}$ satisfies axioms **A1-A4**. If $\underline{P}$ is 2-monotonic and with $\mathcal{F} = \mathcal{F}_{12}$ defined above, then $\mathrm{MMI}_{\mathcal{F}_{12}}$ satisfies axioms **A1-A5**. For **A5**, the subadditivity becomes $\mathrm{MMI}_{\mathcal{F}_{12}} \leq \mathrm{MMI}_{\mathcal{F}_1} + \mathrm{MMI}_{\mathcal{F}_2}$. If the notion of independence in A6 is taken to be strong independence in the sense of Cozman [99], **A6** also holds, and $\mathrm{MMI}_{\mathcal{F}_{12}} = \mathrm{MMI}_{\mathcal{F}_1} + \mathrm{MMI}_{\mathcal{F}_2}$.*

*Proof.* **Axiom A1.** Starting from Axiom A1 that $\mathrm{MMI}_{\mathcal{F}}$ is non-negative and bounded. First, recall that lower probabilities are super-additive, meaning, for $A, B \in \Sigma_{\mathcal{X}}$ with $A \cap B = \emptyset$, we have

$$
\underline{P}(A \cup B) \geq \underline{P}(A) + \underline{P}(B).
$$

Now let $A = \{f \geq t\}$ and $B = \{f < t\}$, then we have

$$
1 \geq \underline{P}(\{f \geq t\}) + \underline{P}(\{f < t\}).
$$

This means the integrand in Equation (3) is always non-negative, thus, $\mathrm{MMI}_{\mathcal{F}}$ is always non-negative. To show boundedness, notice that,

$$
\begin{aligned}
|\mathrm{MMI}_{\mathcal{F}}(\underline{P})| &= \left| \sup_{f \in \mathcal{F}} \int_{\underline{f}}^{\overline{f}} [1 - (\underline{P}(\{f \geq t\}) + \underline{P}(\{f < t\}))] \right| \\
&\leq \sup_{f \in \mathcal{F}} \int_{\underline{f}}^{\overline{f}} |1 - (\underline{P}(\{f \geq t\}) + \underline{P}(\{f < t\}))| \, dt \\
&\leq \sup_{f \in \mathcal{F}} \int_{\underline{f}}^{\overline{f}} 1 \, dt \\
&= \sup_{f \in \mathcal{F}} \sup_{x,y \in \mathcal{X}} |f(x) - f(y)| \\
&\leq 2 \sup_{f \in \mathcal{F}} \sup_{x \in \mathcal{X}} |f(x)|.
\end{aligned}
$$

As $f \in C_b(\mathcal{X})$, by definition, it is a bounded function, thus $\sup_{f \in \mathcal{F}} \sup_{x \in \mathcal{X}} |f(x)|$ is bounded.

**Axiom A2.** For continuity, our goal is to show that given a sequence of lower probabilities $\underline{P}_n$ converges to $\underline{P}$ in the Choquet weak convergence sense, then $\mathrm{MMI}_{\mathcal{F}}(\underline{P}_n) \to \mathrm{MMI}_{\mathcal{F}}(\underline{P})$. Pick $\epsilon > 0$, we know there exists $n_\epsilon \in \mathbb{N}$ such that for all $n > n_\epsilon$,

$$
\left| \oint f d\underline{P}_n - \oint f d\underline{P} \right| < \epsilon
$$

for all $f \in C_b(\mathcal{X})$. An immediate result that follows from Troffaes and De Cooman [19, Proposition C.5.iv] for upper probabilities is,

$$
\left| \oint f d\overline{P}_n - \oint f d\overline{P} \right| = \left| \oint -f d\underline{P}_n - \oint -f d\underline{P} \right| < \epsilon
$$

since $-f \in C_b(\mathcal{X})$. Now with this $n$, we know
$$|\mathrm{MMI}_{\mathcal{F}}(\underline{P}_n) - \mathrm{MMI}_{\mathcal{F}}(\underline{P})|$$

$$= \left| \sup_{f \in \mathcal{F}}\{\oint f d\overline{P}_n - \oint f d\underline{P}_n\} - \sup_{g \in \mathcal{F}}\{\oint g d\overline{P} - \oint g d\underline{P}\} \right|$$

$$= \left| \sup_{f \in \mathcal{F}}\{\oint f d\overline{P}_n - \oint f d\overline{P} + \oint f d\overline{P} - \oint f d\underline{P} + \oint f d\underline{P} - \oint f d\underline{P}_n\} - \sup_{g \in \mathcal{F}}\{\oint g d\overline{P} - \oint g d\underline{P}\} \right|$$

$$\leq \left| \sup_{f \in \mathcal{F}}\{\oint f d\overline{P}_n - \oint f d\overline{P}\} + \sup_{f \in \mathcal{F}}\{\oint f d\underline{P}_\epsilon - \oint f d\underline{P}\} + \sup_{g \in \mathcal{F}}\{\oint g d\overline{P} - \oint g d\underline{P}\} - \sup_{g \in \mathcal{F}}\{\oint g d\overline{P} - \oint g d\underline{P}\} \right|$$

$$\leq \sup_{f \in \mathcal{F}}\{| \oint f d\overline{P}_n - \oint f d\overline{P}| + \sup_{f \in \mathcal{F}}\{| \oint f d\underline{P}_n - \oint f d\underline{P}|\} < 2\epsilon.$$

Thus, we have proven continuity of $\mathrm{MMI}_{\mathcal{F}}$.

**Axiom A3.** To prove monotonicity, notice if $\underline{P}$ is setwise dominated by $\underline{Q}$, then we have for any $t$ and any $f \in \mathcal{F}$,
$$\underline{P}(\{f \geq t\}) + \underline{P}(\{f < t\}) \leq \underline{Q}(\{f \geq t\}) + \underline{Q}(\{f < t\})$$
$$\implies 1 - (\underline{P}(\{f \geq t\}) + \underline{P}(\{f < t\})) \geq 1 - (\underline{Q}(\{f \geq t\}) + \underline{Q}(\{f < t\})).$$
Since the integrand of one integral is always at least as large as the other one, by Troffaes and De Cooman [19, Proposition C.5.vi], we have
$$\int_{\underline{f}}^{\overline{f}} 1 - (\underline{P}(\{f \geq t\}) + \underline{P}(\{f < t\}))dt \geq \int_{\underline{f}}^{\overline{f}} 1 - (\underline{Q}(\{f \geq t\}) + \underline{Q}(\{f < t\}))dt$$
$$\implies \mathrm{MMI}_{\mathcal{F}}(\underline{P}) \geq \mathrm{MMI}_{\mathcal{F}}(\underline{Q}).$$

**Axiom A4.** Showing probability consistency is almost trivial as any $P \in \mathcal{P}(\mathcal{X})$ is self-conjugate, therefore $\mathrm{MMI}_{\mathcal{F}}(P) = 0$.

**Axiom A5.** Recall $\mathcal{F}_{12}$ is defined as
$$\mathcal{F}_{12} := \{f \in C_b(\mathcal{X}) \mid f(x_1, x_2) = f_1(x_1) + f_2(x_2) \text{ for some } f_1 \in C_b(\mathcal{X}_1), f_2 \in C_b(\mathcal{X}_2)\}$$
also that for axiom A5 we are working with 2-monotonic lower probabilities $\underline{P}$, meaning for any event $A, B \in \Sigma_{\mathcal{X}}$,
$$\underline{P}(A \cup B) + \underline{P}(A \cap B) \geq \underline{P}(A) + \underline{P}(B). \tag{4}$$
Now with Troffaes and De Cooman [19, Proposition C.7] we know that for 2-monotone capacities, the Choquet integral is super-additive, meaning, for $f \in \mathcal{F}_{12}$
$$\oint f d\underline{P} = \oint (f_1 + f_2) d\underline{P} \geq \oint f_1 d\underline{P} + \oint f_2 d\underline{P} = \oint f_1 d\underline{P}_1 + \oint f_2 d\underline{P}_2.$$
Notice the last equality holds because $\underline{P}(\{f_1 \geq t\}) = \underline{P}(\{\{x_1 \in \mathcal{X} \mid f_1(x_1) \geq t\} \times \mathcal{X}_2\}) = \underline{P}_1(\{x_1 \in \mathcal{X}_1 \mid f_1(x_1) \geq t\})$. Similarly, we can show that for upper probabilities of 2-monotone lower probabilities, they are 2-alternating, meaning that
$$\oint f d\overline{P} = \oint (f_1 + f_2) d\overline{P} \leq \oint f_1 d\overline{P} + \oint f_2 d\overline{P} = \oint f_1 d\overline{P}_1 + \oint f_2 d\overline{P}_2$$
Now, combine the two results, we have,
$$\mathrm{MMI}_{\mathcal{F}_{12}}(\underline{P})$$
$$= \sup_{f \in \mathcal{F}_{12}} \left\{ \oint f d\overline{P} - \oint f d\underline{P} \right\}$$
$$\leq \sup_{f \in \mathcal{F}_{12}} \left\{ \oint f_1 d\overline{P}_1 + \oint f_2 d\overline{P}_2 - \left( \oint f_1 d\underline{P}_1 + \oint f_2 d\underline{P}_2 \right) \right\}$$
$$\leq \sup_{f \in \mathcal{F}_1} \left\{ \oint f d\overline{P}_1 - \oint f d\underline{P}_1 \right\} + \sup_{f \in \mathcal{F}_2} \left\{ \oint f d\overline{P}_2 - \oint f d\underline{P}_2 \right\}$$
$$= \mathrm{MMI}_{\mathcal{F}_1}(\underline{P}_1) + \mathrm{MMI}_{\mathcal{F}_2}(\underline{P}_2).$$

**Axiom A6.** Now when $\underline{P}_1$ and $\underline{P}_2$ are strongly independent, then the credal set associated to $\underline{P}$, denote as $\mathcal{C}$, is related to the credal set associated to $\underline{P}_1$ and $\underline{P}_2$, denote as $\mathcal{C}_1$ and $\mathcal{C}_2$, as follows:

$$\mathcal{C}_1 := \{P \in \mathcal{P}(\mathcal{X}) \mid P = P_1 \cdot P_2 \text{ for some } P_1 \in \mathcal{C}_1, P_2 \in \mathcal{C}_2\}.$$

To show that the uncertainty is additive when strong independence holds, we use the following representation of the Choquet integral for 2-monotone lower probabilities,

$$\oint f d\underline{P} = \inf_{P \in \mathcal{C}} \int f dP$$

and similarly for 2-alternating upper probabilities, we have

$$\oint f d\overline{P} = \sup_{P \in \mathcal{C}} \int f dP.$$

Now we can show the result, starting with

$\mathrm{MMI}_{\mathcal{F}_{12}}(\underline{P})$

$$= \sup_{f \in \mathcal{F}_{12}} \left\{ \oint f d\overline{P} - \oint f d\underline{P} \right\}$$

$$\overset{(A)}{=} \sup_{f_1 \in \mathcal{F}_1, f_2 \in \mathcal{F}_2} \left\{ \sup_{P \in \mathcal{C}} \int (f_1 + f_2) dP - \inf_{Q \in \mathcal{C}} \int (f_1 + f_2) dQ \right\}$$

$$\overset{(B)}{=} \sup_{f_1 \in \mathcal{F}_1, f_2 \in \mathcal{F}_2} \left\{ \sup_{P_1 \in \mathcal{C}_1, P_2 \in \mathcal{C}_2} \left\{ \int f_1 dP_1 + \int f_2 dP_2 \right\} - \inf_{Q_1 \in \mathcal{C}_1, Q_2 \in \mathcal{C}_2} \left\{ \int f_1 dQ_1 - \int f_2 dQ_2 \right\} \right\}$$

$$= \sup_{f_1 \in \mathcal{F}_1, f_2 \in \mathcal{F}_2} \left\{ \sup_{P_1 \in \mathcal{C}_1} \int f_1 dP_1 + \sup_{P_2 \in \mathcal{C}_2} \int f_2 dP_2 - \inf_{Q_1 \in \mathcal{C}_1} \int f_1 dQ_1 - \inf_{Q_2 \in \mathcal{C}_2} \int f_2 dQ_2 \right\}$$

$$\overset{(C)}{=} \sup_{f_1 \in \mathcal{F}_1, f_2 \in \mathcal{F}_2} \left\{ \oint f_1 d\overline{P_1} + \oint f_2 d\overline{P_2} - \oint f_1 d\underline{P_1} - \oint f_2 d\underline{P_2} \right\}$$

$$= \sup_{f_1 \in \mathcal{F}_1} \left\{ \oint f_1 d\overline{P_1} - \oint f_1 d\underline{P_1} \right\} + \sup_{f_2 \in \mathcal{F}_2} \left\{ \oint f_2 d\overline{P_2} - \oint f_2 d\underline{P_2} \right\}$$

$$= \mathrm{MMI}_{\mathcal{F}_1}(\underline{P}_1) + \mathrm{MMI}_{\mathcal{F}_2}(\underline{P}_2).$$

In step (A), we used the fact that the Choquet integral of 2-monotone capacities is the lower envelope of the set of expectations with respect to the credal set [145, Lemma 2]. In Step (B), we used the fact that due to strong independence, every element in $P \in C$ can be written as a product of some $P_1 \in \mathcal{C}_1$ and $P_2 \in \mathcal{C}_2$ and the fact that Lebesgue integration is linear, and $\int f_1(x_1) P(dx_1, dx_2) = \int f_1(x_1) P_1(dx_1)$. Finally, in step (C), we used Lemma 24, which stated that marginals of a 2-monotone capacity remain 2-monotone, therefore, we can write $\inf_{P_1 \in \mathcal{C}_1} \int f_1 dP_1$ back as a Choquet integral. $\qquad \square$

## B.13 Proof of Proposition 21

Finally, to prove the result for the upper bound, we first recall an intermediate result from Montes et al. [146, Proposition 8], which provides a construction for the best pessimistic $\epsilon$-contamination set approximation to any lower probability $\underline{P} \in \mathcal{P}(\mathcal{X})$. Understanding this proposition requires clarifying the concept of dominance and outer approximation.

**Definition 25** (Outer approximation). *Let $\mathcal{X}$ be finite and $\underline{Q}, \underline{P} \in \mathcal{P}(\mathcal{X})$ two lower probabilities. We say $\underline{Q}$ is an outer approximation of $\underline{P}$ if for every event $A \in 2^{\mathcal{X}}$, $\underline{Q}(A) \leq \underline{P}(A)$.*

The direction of the inequality might be confusing at first, but by realising $\underline{Q}(A) \leq \underline{P}(A)$ for every event $A \in 2^{\mathcal{X}}$, it means the core of $\underline{Q}$, i.e.

$$\mathcal{M}(\underline{Q}) = \{Q \in \mathbb{P}(\mathcal{X}) : Q(A) \geq \underline{Q}(A) \quad \forall A \in 2^{\mathcal{X}}\},$$

is at least as large as the core of $\underline{P}$, i.e. $\mathcal{M}(\underline{P}) \subseteq \mathcal{M}(\underline{Q})$.

**Definition 26** (Undominated outer approximation in $\mathcal{C}_\star(\mathcal{X})$)**.** *Let $\underline{P} \in \underline{\mathcal{P}}(\mathcal{X})$ be a lower probability. Let $\mathcal{C}_\star(\mathcal{X}) \subseteq \underline{\mathcal{P}}(\mathcal{X})$ be a subspace of which the outer approximation $\underline{Q}$ resides. We say $\underline{Q}$ is an undominated outer approximation of $\underline{P}$ in $\mathcal{C}_\star(\mathcal{X})$ if there exists no other lower probabilities $\underline{Q}'$ in $\mathcal{C}_\star(\mathcal{X})$ such that $\mathcal{M}(\underline{P}) \subseteq \mathcal{M}(\underline{Q}') \subsetneq \mathcal{M}(\underline{Q})$.*

Now we have all the language to understand the result from Montes et al. [146]. Let $\mathcal{C}_\epsilon(\mathcal{X}))$ denote the space of all $\epsilon$-contamination models defined on $\mathcal{X}$.

**Proposition 27.** *Let $\underline{P} \in \underline{\mathcal{P}}(\mathcal{X})$ be a lower probability. Define $\epsilon \in (0,1)$ and the probability $P_0$ by:*

$$\epsilon = 1 - \sum_{x \in \mathcal{X}} \underline{P}(\{x\}), \qquad P_0(\{x\}) = \frac{\underline{P}(\{x\})}{\sum_{x \in \mathcal{X}} \underline{P}(\{x\})}, \text{ for every } x \in \mathcal{X}.$$

*Denote by $\underline{P}_\epsilon$ the $\epsilon$-contaminated model constructed following Lemma 15. Then, $\underline{P}_\epsilon$ is the unique undominated outer approximation of $\underline{P}$ in $\mathcal{C}_\epsilon(\mathcal{X})$.*

With this result, we can now derive the upper bound for $\mathrm{MMI}_{\mathcal{F}_{TV}}(\underline{P})$.

**Proposition 21.** *Let $\mathcal{X}$ be finite. For any $\underline{P} \in \underline{\mathcal{P}}(\mathcal{X})$, $\mathrm{MMI}_{\mathcal{F}_{TV}}(\underline{P}) \leq 1 - \sum_{x \in \mathcal{X}} \underline{P}(\{x\})$.*

*Proof.* We know for any $\underline{P}$, $\underline{P}_\epsilon$ constructed as in Proposition 27, is the unique undominated outer approximation of $\underline{P} \in \mathcal{C}_\epsilon(\mathcal{X})$. Since it is an outer approximation, meaning that $\underline{P}_\epsilon(A) \leq \underline{P}(A)$ for all events $A \in 2^{\mathcal{X}}$, therefore by the monotonicity axiom (Axiom 3) in theorem 20, we know for any $\mathcal{F} \subseteq C_b(\mathcal{X})$, we have

$$\mathrm{MMI}_{\mathcal{F}}(\underline{P}) \leq \mathrm{MMI}_{\mathcal{F}}(\underline{P}_\epsilon).$$

Now choose $\mathcal{F} = \mathcal{F}_{TV}$ as in Definition 13, then along with Proposition 19, we have

$$\mathrm{MMI}_{\mathcal{F}_{TV}}(\underline{P}) \leq \mathrm{MMI}_{\mathcal{F}_{TV}}(\underline{P}_\epsilon)$$
$$= \epsilon.$$
$$= 1 - \sum_{x \in \mathcal{X}} \underline{P}(\{x\}).$$

This concludes the derivation. $\qquad\square$

## C  Further experimental details

### C.1  Ablation study: Correlation with Generalised Hartley and Entropy Difference.

In the main text, we observed that the performance of MMI, its linear-time upper bound MMI-Lin, and the generalised Hartley measure are nearly equivalent in the context of selective classification. This raises the question: *are they numerically equivalent?* That is, is generalised Hartley and MMI using the total variation function space, i.e. lower TV, providing the same value? While staring at the equations tells us they are not equivalent, our ablation study suggests that even though they are highly correlated, they are not the same.

To investigate that, we use the UCI wine dataset [148], perform one round of train-test split, train 10 random forests models as described in the main text to construct the lower probabilities, and then compute the epistemic uncertainty measurements for the test data. After that, we plot all possible pairwise comparison plots between the methods, i.e.MMI, MMI-Lin, Generalised Hartley, and Entropy difference. We see that MMI and GH are highly correlated but not exactly perfectly correlated. The upper bound is also highly correlated, suggesting empirically (along with experiments in the main text), that this approximation is quite tight. Also, we see that MMI and entropy differences do not correlate that well, which explains the difference in downstream performance in the experiments.

### C.2  Overview of Generalised Hartley measures and Entropy Differences

We refer the reader to the recent survey by Hoarau et al. [123] on this topic. We hereby provide background on the two methods we compared against in the main text.

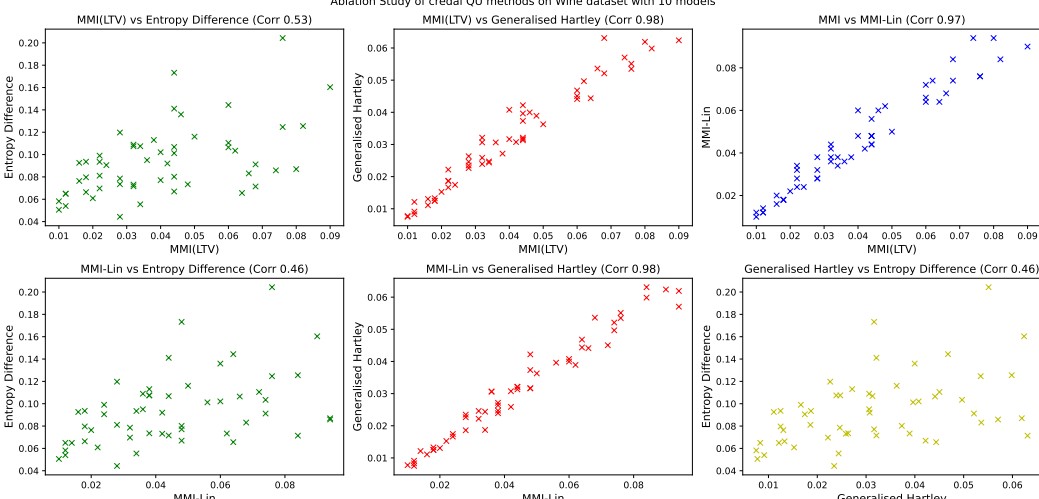

**Figure 3:** Comparing the UQ measurements on the withheld test set of the UCI wine dataset [148]. We see that MMI and GH are highly correlated, but not exactly perfectly correlated. The upper bound is also highly correlated, suggesting empirically (along with experiments in the main text), that this approximation is quite tight. Finally, we see that MMI and entropy differences do not correlate that well.

**Generalised Hartley Measure.** Generalised Hartley measure, as the name suggests, is a generalisation of the classical Hartley measure, which is defined as follows:

**Definition 28** (Hartley Measure). *Let $\mathcal{X}$ be finite. A Hartley measure $U_H$ is a function $2^{\mathcal{X}} \mapsto \mathbb{R}$ such that $U_H(A) = \log_2 |A|$ for $A \in 2^{\mathcal{X}}$.*

The Hartley measure can thus be understood as a measure of uncertainty over sets, which can also be viewed as a function on natural numbers. While the expression is simple, Rényi showed that the Hartley measure is the only function mapping natural numbers to the reals that satisfies:

1. $U_H(mn) = U_H(m) + U_H(n)$    (additivity)
2. $U_H(m) \geq U_H(m+1)$    (monotonicity)
3. $U_H(2) = 1$    (normalisation)

These are natural conditions for measuring the amount of uncertainty, or equivalently, information, within a set. The generalised Hartley measure extends this idea of measuring uncertainty in sets in the following way,

**Definition 29** (Generalised Hartley [94]). *Let $\mathcal{X}$ be finite. Given a lower probability $\underline{P} \in \mathcal{P}(\mathcal{X})$, the generalised Hartley $U_{GH}$ maps $\underline{P}$ to $\mathbb{R}$ as follows:*

$$U_{GH}(\underline{P}) = \sum_{A \subseteq \mathcal{X}} m_{\underline{P}}(A) \log_2(|A|),$$

*where the mass function $m_{\underline{P}} : 2^{\mathcal{X}} \to [0,1]$ is the Möbius inverse of $\underline{P}$, defined as,*

$$m_{\underline{P}}(A) = \sum_{B \subseteq A} (-1)^{(|A|-|B|)} \underline{P}(B).$$

It is well known that the Möbius inverse is an equivalent representation of a lower probability, in the sense that, once the mass function is known, we can recover $\underline{P}$ by computing,

$$\underline{P}(A) = \sum_{B \subseteq A} m(B).$$

**Entropy Differences.** To measure aleatoric uncertainty for a given distribution $P$, the Shannon entropy is an intuitive solution. The Shannon entropy, or simply entropy, measures the amount of information contained or provided by a source of information.

**Definition 30** (Shannon Entropy). *Let $\mathcal{X}$ be finite. The Shannon entropy of a distribution $P$ is measured by*

$$U_{Sh}(P) = -\sum_{x \in \mathcal{X}} P(\{x\}) \log_2(P(\{x\})).$$

Now, when given a set of probabilities, i.e. a credal set, to measure the amount of epistemic uncertainty, or in some literature, they call this non-specificity, one natural approach would be to measure the largest difference between the entropies of any two distributions within the set. Formalised below,

**Definition 31** (Entropy Difference). *Let $\mathcal{X}$ be finite and $\mathcal{C}$ a credal set. The entropy difference is measured by,*

$$\max_{P \in \mathcal{C}} U_{Sh}(P) - \min_{Q \in \mathcal{C}} U_{Sh}(Q).$$

This measure seems intuitive, but violates the monotonicity axioms that a sensible credal UQ measure should satisfy. Consider a credal set $\mathcal{C}$ and $P_1, P_2$ the distributions that attained the maximum and minimum of the entropies of the distributions in $\mathcal{C}$. Now, enlarge $\mathcal{C}$ to $\mathcal{C}'$ by adding distributions that have strictly less entropy than $P_1$ but more entropy than $P_2$, then we have $\mathcal{C} \subset \mathcal{C}'$, but the entropy differences will stay the same. This violates the monotonicity axioms that say, when a credal set is strictly larger than the other, then the former should be deemed more epistemically uncertain than the latter.

Nonetheless, the entropy difference is still often used in practice as it is simple to compute, and it can be used to decompose 'total uncertainty' into 'aleatoric' and 'epistemic' components. See for example in Abellán et al. [100], the author defined,

$$\underbrace{\max_{P \in \mathcal{C}} U_{Sh}(P)}_{\text{Total Uncertainty}} = \underbrace{\min_{Q \in \mathcal{C}} U_{Sh}(Q)}_{\text{Aleatoric Uncertainty}} + \underbrace{\max_{P \in \mathcal{C}} U_{Sh}(P) - \min_{Q \in \mathcal{C}} U_{Sh}(Q)}_{\text{Epistemic Uncertainty}}.$$

### C.3 Implementation details

We provide full experimental details here, which were abbreviated in Section 6 due to space constraints. The experiments were executed on a machine with 8 vCPUs, 30 GB memory, with a NVIDIA V100 GPU.

We evaluate the performance of Maximum Mean Imprecision using a selective classification task, following the setup in Shaker and Hüllermeier [57] and Shaker and Hüllermeier [80]. Each dataset is split into training and test sets. For tabular datasets (the Obesity dataset from UCI and Digits dataset from Sci-kit learn), we train 10 random forests with randomly chosen hyperparameters (e.g., tree depth) on the same training set, and evaluate them on the test set. For image datasets (CIFAR10 and CIFAR100), we use 10 pretrained neural networks per task, available at https://github.com/chenyaofo/pytorch-cifar-models. These models were trained using PyTorch's default CIFAR training sets, and we evaluate the standard CIFAR test sets by dividing them into 10 buckets to introduce variability.

For both tabular and image data, we use the centroid of the credal set as the predictor, similar to standard ensemble methods. The corresponding lower probability is computed by evaluating the most pessimistic likelihood across the set of predictions for each possible outcome.

## D   IIPM with epsilon-contamination set

In this section of the appendix, we continue from Section 3 and dive deeper into the connections between IIPM and the epsilon contamination model—a popular class of imprecise models.

### D.1 Lower Probability Kantorovich Problem

In this subsection, we provide the background on a recently considered problem called the lower probability Kantorovich problem, proposed in Caprio [56]. We start by reviewing what a classical Kantorovich problem is.

**Classical Kantorovich problem.** Let $P, Q \in \mathcal{P}(\mathcal{X})$ be two probability measures on $\mathcal{X}$. Let $c : \mathcal{X} \times \mathcal{X} \to \mathbb{R}_+$ be a measurable cost function that gives us the cost of moving on unit of probability mass from the first argument to the other. Then the classical Kantorovich problem is the following optimisation problem,

$$\arg\inf_{\alpha \in \Gamma(P,Q)} \left\{ \int c(x, z) d\alpha(x, z) \right\}$$

where $\Gamma(P, Q)$ is the set of all joint probability measures whose marginals are $P$ and $Q$. $\alpha$ is also denote as the *transportation plan*, and this is also famously known as the *optimal transport* problem.

**Lower Probability Kantorovich problem.** Caprio [56] consider the following research question,

*What does Kanotorovich's problem look like, when instead of transporting probability measures, we transport lower probabilities?*

As his answer, Caprio [56] provided the following characterisation of the problem,

**Definition 32** (Lower Probability Kantorovich's OT problem; LPK). *Let $c : \mathcal{X} \times \mathcal{X} \to \mathbb{R}_+$ be a Borel measurable cost function. Given lower probabilities $\underline{P}$ and $\underline{Q}$ on $\mathcal{X}$, we want to find the joint lower probability $\underline{alpha}$ on $\mathcal{X} \times \mathcal{X}$ that solves the following optimisation problem*

$$\arg\inf_{\underline{\alpha} \in \Gamma(\underline{P},\underline{Q})} \left\{ \int_{\mathcal{X} \times \mathcal{X}} c(x, z) d\underline{\alpha}(x, z) \right\},$$

*where $\Gamma(\underline{P}, \underline{Q})$ is the collection of all joint lower probabilities on $\mathcal{X} \times \mathcal{X}$ whose marginals on $\mathcal{X}$ are $\underline{P}$ and $\underline{Q}$, respectively.*

While this might seem like a straightforward extension from the classical formulation, it is important to note that for imprecise probability theory, there is no unique way to perform conditioning, which means extra care has to be taken into defining $\Gamma(\underline{P}, \underline{Q})$. In particular, they focus on a subset of the joint lower probabilities constructed from using geometric conditioning, and called the corresponding LPK problem restricted to such conditioning set the restricted LPK problem.

Later on in Theorem 11 of [56], they managed to show that the restricted LPK problem coincides exactly with the classical Kantorovich for $epsilon$-contaminated sets. We managed to recover this result in our Theorem 16 without needing to consider any specific type of conditioning. In the future, we will investigate how could our result complements to their theory, perhaps allow them to consider other types of conditioning operations in IP.

### D.2 Nonparametric Estimator of IIPM with $\epsilon$-contamination set using kernel distance.

Now consider $\epsilon, \delta \in (0, 1)$ two contamination levels, and distributions $P, Q \in \mathcal{P}(\mathcal{X})$ which we have i.i.d samples from. Specifically, let $X_1, \ldots, X_n \overset{iid}{\sim} P$ and $Z_1, \ldots, Z_m \overset{iid}{\sim} Q$ be random variables taking values in $\mathcal{X}$. We are interested in quantifying the difference between the $\epsilon$-contaminated model of $P$, i.e. $\underline{P}_\epsilon$, with respect to the $\delta$-contaminated model $Q$, i.e. $\underline{Q}_\delta$.

**A short overview of kernel distances (MMD).** For generic spaces $\mathcal{X}$, with iid samples from $P$ and $Q$, a popular class of non-parametric discrepancy estimator is the maximum mean discrepancy (MMD) [67, 71]. Specifically, pick a kernel function $k : \mathcal{X} \times \mathcal{X} \to \mathbb{R}$ and consider the uniform ball in the corresponding reproducing kernel Hilbert space (RKHS), i.e. $\mathcal{F}_k = \{f \in \mathcal{H}_k \text{ s.t. } \|f\|_k = 1\}$, where $\| \cdot \|_k$ stands for the RKHS norm. The MMD can be expressed as an IPM with respect to

function class $\mathcal{F}_k$,

$$\mathrm{MMD}_k(P,Q) = \mathrm{IPM}_{\mathcal{F}_k}(P,Q)$$

$$= \sup_{f \in \mathcal{H}_k; \|f\|_k = 1} \left\{ \left| \int f dP - \int f dQ \right| \right\}$$

$$\overset{(A)}{=} \sup_{f \in \mathcal{H}_k; \|f\|_k = 1} \left\{ \left| \left\langle f, \int k(X,\cdot)dP(X) \right\rangle - \left\langle f, \int k(X,\cdot)dQ(X) \right\rangle \right| \right\}$$

$$= \sup_{f \in \mathcal{H}_k; \|f\|_k = 1} \left\{ \left| \left\langle f, \int k(X,\cdot)dP(X) - \int k(X,\cdot)dQ(X) \right\rangle \right| \right\}$$

$$\overset{(B)}{=} \left\| \int k(X,\cdot)dP(X) - \int k(X,\cdot)dQ(X) \right\|_k$$

$$\overset{(C)}{=} \|\mu_P - \mu_Q\|_k$$

where in step (A) we first use the reproducing property of RKHS functions, ($f(x) = \langle f, k(x,\cdot)\rangle$), and then use the linearity of the inner product to 'push' the expectation inside. In step (B) we use the fact that inner product is maximised when the two vectors align, so we pick the unit-norm function to be

$$f = \frac{\int f(k(X,\cdot)dP(X) - \int k(X,\cdot)dQ(X)}{\|\int f(k(X,\cdot)dP(X) - \int k(X,\cdot)dQ(X)\|}.$$

Finally, in step (C), we simply write the (Bochner) integral of the feature representation into a more familiar-looking expression, called the kernel mean embedding [149, 150]. This simple expression facilitates further simplification, i.e.

$$\mathrm{MMD}_k(P,Q)^2 = \|\mu_P - \mu_Q\|_k^2$$

$$= \langle \mu_P - \mu_Q, \mu_P - \mu_Q \rangle$$

$$\overset{(D)}{=} \mathbb{E}_{X,X'\sim P}[k(X,X')] - 2\mathbb{E}_{X\sim P, Z\sim Q}[k(X,Z)] + \mathbb{E}_{Z,Z'\sim Q}[k(Z,Z')].$$

Meaning that we don't need a parametric assumption on how the data is distributed, as long as we have a way to define similarity between instances through a kernel function, we can estimate the difference between the distributions based on their samples by estimating the expectations in step D. Furthermore, for a wide class of kernels, known as charactersitic kernels, the mapping from $P \mapsto \int k(X,\cdot)dP(X)$ is injective, thus $\mathrm{MMD}_k$ is a proper metric between probability distributions. Owing to its simplicity, kernel mean embeddings and MMDs have been used to tackle a broad range of statistical tasks, ranging from hypothesis testing [71] to parameter estimation [151, 152], causal inference [153, 154], feature attribution [155, 43], and learning on distributions [156, 157].

**Generalising to contamination sets.** In general, given a lower probability $\underline{P}$, the notion of samples from $\underline{P}$ is ill-defined as lower probability often is used to encode subjective assessment rather than describing the data-generating process. As such, devising a sample-based estimator for the Choquet integral, akin to Monte Carlo estimation for the Lebesgue integral, is not yet possible. Nonetheless, in the case of utilising lower probability constructed through an epsilon contamination model, this is possible.

Recall $\epsilon, \delta \in (0,1)$ are two contamination level, with $\underline{P}_\epsilon$ and $\underline{Q}_\delta$ the corresponding contaminated models. We are now interested in quantifying the difference between this two lower probabilities using the IIPM framework through the kernel distances. As before, pick $\mathcal{F}_k = \{f \in \mathcal{H}_k \text{ s.t. } \|f\|_k = 1\}$, then we have

$$\mathrm{IIPM}_{\mathcal{F}_k}(\underline{P}_\epsilon, \underline{Q}_\delta) = \sup_{f \in \mathcal{H}_k; \|f\|_k = 1} \left\{ \left| \oint f d\underline{P}_\epsilon - \oint f d\underline{Q}_\delta \right| \right\}$$

$$\overset{(i)}{=} \sup_{f \in \mathcal{H}_k; \|f\|_k = 1} \left\{ \left| \int_{\underline{f}}^{\overline{f}} \left( \underline{P}_\epsilon(\{f > t\}) - \underline{Q}_\delta(\{f > t\}) \right) dt \right| \right\}$$

$$\overset{(ii)}{=} \sup_{f \in \mathcal{H}_k; \|f\|_k = 1} \left\{ \left| (1-\epsilon)\int f dP - (1-\delta)\int f dQ \right| \right\}$$

$$\overset{(iii)}{=} \|(1-\epsilon)\mu_P - (1-\delta)\mu_Q\|_k$$

where in step (i) we simply expand our the definition of Choquet integral. In step (ii), we follow Lemma 15 and the proof of Theorem 16 to express the Choquet integral now as a weighted Lebesgue integral. In step (iii), we follow the derivations of standard MMD provided in the previous paragraph.

Subsequently, the square of $\text{IIPM}_{\mathcal{F}_k}(\underline{P}_\epsilon, \underline{Q}_\delta)$ can be expressed as,

$$\begin{aligned}
\text{IIPM}_{\mathcal{F}_k}(\underline{P}_\epsilon, \underline{Q}_\delta) = (1-\epsilon)^2 \mathbb{E}_{X,X'\sim P}[k(X,X')] \\
- 2(1-\epsilon)(1-\delta)\mathbb{E}_{X\sim P, Z\sim Q}[k(X,Z)] + (1-\delta)^2 \mathbb{E}_{Z,Z'\sim Q}[k(Z,Z')].
\end{aligned}$$

This allows us to then construct a non-parametric (unbiased) estimator of (the square of )$\text{IIPM}_{\mathcal{F}_k}(\underline{P}_\epsilon, \underline{Q}_\delta)$ as follows,

$$\begin{aligned}
\widehat{\text{IIPM}^2_{\mathcal{F}_k}}(\underline{P}_\epsilon, \underline{Q}_\delta) = (1-\epsilon)^2 \frac{1}{n(n-1)} \sum_{i=1}^n \sum_{j=1,j\neq i}^n k(X_i, X_j) \\
- 2(1-\epsilon)(1-\delta)\frac{1}{nm} \sum_{i=1}^n \sum_{j=1}^m k(X_i, Z_j) + (1-\delta)^2 \sum_{i=1}^m \sum_{j=1,j\neq i}^m k(Z_i, Z_j)
\end{aligned}$$

Due to project scope, we did not further investigate the concrete applications of such a non-parametric worst-case probability discrepancy estimator, but in future work, we will explore its use case in robust two-sample testing, akin to Schrab and Kim [158], or in generative model training, akin to Li et al. [131].

