# OpenReview forum: "Integral Imprecise Probability Metrics"
_NeurIPS.cc/2025/Conference — NeurIPS 2025 poster_

### Official Review · Reviewer_6KfT · 2025-06-19

**Clarity:** 2
**Significance:** 2
**Originality:** 3
**Rating:** 4
**Confidence:** 2

**Summary:**

This paper introduces imprecise integral probability metrics. Such metrics are a generalization of classical integral probability metrics to compare two distributions, such as the Wasserstein and total variation distance. The imprecise integral probability metrics allow to define a distance over capacities instead of probability measures. As a second contribution, the new metric not only allows to compare two imprecise probability models, but also to quantify epistemic uncertainty in a single imprecise probability model, using the notion of maximum mean imprecision (MMI). In a selective classification experiment that involves computing classification-rejection curves on two UCI and tow image benchmark datasets, the MMI is compared to two baselines: the entropy-based approach to quantify epistemic uncertainty and the generalized Heartley measure.

**Questions:**

See above.

**Ethical Concerns:**

["NO or VERY MINOR ethics concerns only"]

**Final Justification:**

I thank the authors for a detailed response to my comments. In general I remain on the positive side for this paper.

**Limitations:**

Limitations are briefly discussed in Section 7.

**Paper Formatting Concerns:**

More experimental details should be discussed in the main paper.

**Quality:**

3

**Strengths And Weaknesses:**

My knowledge of imprecise probability theory is very limited, so I find it hard to give constructive feedback on this paper.

Despite my limited knowledge on the topic, I could follow Sections 3 and 4, which constitute the theoretical contributions of this paper, quite well. The authors are able to discuss the novel concepts in a mathematically rigorous manner, which is appreciated by this reviewer. The authors also give evidence of a very good knowledge of the literature on imprecise probability theory (the paper contains 135 references!). To the best of my knowledge, the introduced concepts are also novel.

The discussion is less clear in Section 6, where the application of the new concepts for ML is showcased on benchmark datasets. That section is very short and some essential information is missing. First of all, how is epistemic uncertainty computed for the baselines in the setting that you consider? I guess that, for the entropy-based method, this is similar to reference [77]. For the generalized Heartley measure, do you consider the volume of the convex hull formed by the ensemble members? For the method presented by the authors, it is also not clear to me how epistemic uncertainty is computed. How do you relate the formulas in Definition 17 or Proposition 18 to the convex hull of the ensemble members? This might be discussed in appendices, but it should be discussed in the main paper (sorry, I have to review 6 papers for Neurips, so I don't have time to read appendices).

In the experimental setup, it was not so clear to me why the authors only focus on epistemic uncertainty. One can also reject certain instances because of aleatoric uncertainty. That would be a different type of rejection, but it would be interesting to see how this relates to epistemic uncertainty.

Finally, it is a subjective opinion, but I am not sure whether Neurips is a good venue for this type of work. The key contributions are mainly in imprecise probability theory and most references in the paper also come from that field. Furthermore, only the contributions of Section 4 are applicable to ML models and the experiment in Section 6 is quite small.

---

> ### Author Rebuttal · Authors · 2025-07-27
>
> We thank the reviewer for their thoughtful and genuine comments about the clarity of the paper. We address your concerns below:
>
> ------
>
> **Q1: Discussion is less clear in Section 6.**
>
> A1: We thank the reviewer for pointing out the relative lack of depth in the discussion of Section 6 compared to the theoretical and methodological content in Sections 3 and 4. As correctly noted, we have provided extensive additional experimental details in Appendix C. Specifically, Section C.1 presents an ablation study comparing the three credal-based uncertainty measures; Section C.2 offers a detailed exposition on the Generalised Hartley and entropy differences; and Section C.3 describes the full experimental setup used in the main text.
>
> While we would have liked to include these details in the main body, we made a conscious decision to prioritise the clarity and accessibility of our theoretical contributions—particularly the new theorems and propositions. Given that the experimental framework for evaluating epistemic uncertainty quantification via selective classification is well-established and used in prior work, we opted to place the implementation specifics in the appendix to maintain a focused narrative in the main text.
>
> As we will have one extra page in the camera ready, we will move some of the details in Appendix C.3 to the main text to make the main text more self-contained.
>
> ------
>
> **Q2: “In the experimental setup, it was not so clear to me why the authors only focus on epistemic uncertainty. One can also reject certain instances because of aleatoric uncertainty. That would be a different type of rejection, but it would be interesting to see how this relates to epistemic uncertainty.”**
>
> A2: We agree it is interesting to check whether we should reject instances based on low EU or high AU; However, this experiment is not too relevant to the main purpose of this paper: which is introducing a new metric for IP models and to quantify the amount of imprecision (EU) therein.
>
> Also, this particular research question has been studied in [2], where the authors empirically demonstrate that it is generally better to consider EU rather than AU in these selective classification problems. We think this is case-specific, as logically if you have a label with very high AU, there is little signal for one to make an accurate prediction anyway, so we should defer making a prediction at this instance.
>
> [2] Nguyen, Vu-Linh, Sébastien Destercke, and Eyke Hüllermeier. "Epistemic uncertainty sampling." International Conference on Discovery Science. Cham: Springer International Publishing, 2019.
>
> ------
>
> **Q3: Finally, it is a subjective opinion, but I am not sure whether Neurips is a good venue for this type of work.**
>
> A3: The reviewer’s concern is understandable. While imprecise probabilities (IP) are not yet a mainstream topic in the machine learning community, their foundational framework for modeling ambiguity and imprecision offers principled tools that can be adapted to address challenges in modern machine learning. This is evidenced by a growing number of IP-related papers published at top ML venues, including ICLR, AISTATS, ICML, and NeurIPS. We surveyed several such works from 2023 to 2025, spanning various IP frameworks (e.g., credal sets, belief functions, upper/lower probabilities), and listed them below. Our proposed IIPM and MMI methods are, in principle, compatible with these frameworks. We hope this provides some evidence that our contributions to IP-based metrics and uncertainty quantification are timely and relevant to the broader ML community.
>
> 1. Sale, Yusuf, Viktor Bengs, Michele Caprio, and Eyke Hüllermeier. "Second-order uncertainty quantification: a distance-based approach." In Proceedings of the 41st International Conference on Machine Learning, pp. 43060-43076. 2024.
> 2. Chau, Siu Lun, Antonin Schrab, Arthur Gretton, Dino Sejdinovic, and Krikamol Muandet. "Credal Two-Sample Tests of Epistemic Uncertainty." In International Conference on Artificial Intelligence and Statistics, pp. 127-135. PMLR, 2025.
> 3. Singh, A., Chau, S.L., Bouabid, S.; Muandet, K.. (2024). Domain Generalisation via Imprecise Learning. ICML 2024
> 4. Zhou, Zi-Hao, Jun-Jie Wang, Tong Wei, and Min-Ling Zhang. "Weakly-Supervised Contrastive Learning for Imprecise Class Labels." ICML 2025
> 5. Javanmardi, A., Stutz, D., & Hüllermeier, E. (2024). Conformalized credal set predictors. Advances in Neural Information Processing Systems, 37, 116987-117014.
> 6. Marinescu, Radu, Junkyu Lee, Debarun Bhattacharjya, Fabio Cozman, and Alexander Gray. "Abductive reasoning in logical credal networks." Advances in Neural Information Processing Systems 37 (2024): 67968-67986.
> 7. Wang, Kaizheng, Fabio Cuzzolin, Keivan Shariatmadar, David Moens, and Hans Hallez. "Credal deep ensembles for uncertainty quantification." Advances in Neural Information Processing Systems 37 (2024): 79540-79572.
> 8. Caprio, Michele, Maryam Sultana, Eleni Elia, and Fabio Cuzzolin. "Credal learning theory." Advances in Neural Information Processing Systems 37 (2024): 38665-38694.
> 9. Eberle, Franziska, Felix Hommelsheim, Alexander Lindermayr, Zhenwei Liu, Nicole Megow, and Jens Schlöter. "Accelerating matroid optimization through fast imprecise oracles." Advances in Neural Information Processing Systems 37 (2024): 93790-93825.
> 10. Chen, Hao, Ankit Shah, Jindong Wang, Ran Tao, Yidong Wang, Xiang Li, Xing Xie, Masashi Sugiyama, Rita Singh, and Bhiksha Raj. "Imprecise label learning: A unified framework for learning with various imprecise label configurations." Advances in Neural Information Processing Systems 37 (2024): 59621-59654.
> 11. Li, Tongxin, et al. "Safe exploitative play with untrusted type beliefs." Advances in Neural Information Processing Systems 37 (2024): 72901-72932.
> 12. Shai, Adam, Lucas Teixeira, Alexander Oldenziel, Sarah Marzen, and Paul Riechers. "Transformers represent belief state geometry in their residual stream." Advances in Neural Information Processing Systems 37 (2024): 75012-75034.
> 13. Bramblett, Daniel, and Siddharth Srivastava. "Belief-state query policies for user-aligned POMDPs." Advances in Neural Information Processing Systems 37 (2024): 52776-52805.
> 14. Wang, Kaizheng, Fabio Cuzzolin, Keivan Shariatmadar, David Moens, and Hans Hallez. "Credal Wrapper of Model Averaging for Uncertainty Estimation in Classification." In The Thirteenth International Conference on Learning Representations.
> 15. Hu, E.S., Ahn, K., Liu, Q., Xu, H., Tomar, M., Langford, A., Jayaraman, D., Lamb, A. and Langford, J., The Belief State Transformer. In The Thirteenth International Conference on Learning Representations.
> 16. Avalos, Raphaël, Florent Delgrange, Ann Nowe, Guillermo Perez, and Diederik M. Roijers. "The Wasserstein Believer: Learning Belief Updates for Partially Observable Environments through Reliable Latent Space Models." In The Twelfth International Conference on Learning Representations.
>
> --------
>
> We hope our responses have added clarity regarding the presentation and the relevance of our work to the broader ML community. We kindly ask the reviewer to possibly reconsider their score in light of our discussion. Thank you for your consideration.

---

### Official Review · Reviewer_ph2P · 2025-07-02

**Clarity:** 3
**Significance:** 3
**Originality:** 3
**Rating:** 4
**Confidence:** 4

**Summary:**

In this paper, the authors propose to a family of statistical distances for imprecise probabilities, as well as the framework Integral Imprecise Probability Metrics (IIPM). Specifically, it leverages the Choquet integral to generalize classical integral probability metrics, supporting comparison across different IP models and quantifying epistemic uncertainty within a single IP model. The authors also provide the theoretical basis with a form of weak convergence of capacities. Based on IIPM, a specific uncertainty measure, Maximum Mean Imprecise (MMI), can be derived to quantify epistemic uncertainty in classification tasks. Empirically, the authors validate MMI with experiments on several benchmark datasets and demonstrate that MMI outperforms baseline uncertainty quantification methods.

**Questions:**

Q1: As mentioned in Section 7, a key limitation is the lack of a sampling-based estimation method for IIPMs. Would there any future plan to address this?

Q2: What is the trade-off and the key of using the efficient upper bound (or `MMI-Lin`)? It would be great to have some elaboration.

Q3: Would the proposed framework be able to be applied to other ML tasks like regression and ranking? If so, it would also be great to have more experiments to validate.

Q4: There are some studies like [a] to leverage uncertainty measures to improve the model quality. Can the proposed framework also do that? It would also be great to have some studies and discussions about this direction.


[a] Jiang, J. Y., Chang, W. C., Zhang, J., Hsieh, C. J., & Yu, H. F. (2023, July). Uncertainty Quantification for Extreme Classification. In Proceedings of the 46th International ACM SIGIR Conference on Research and Development in Information Retrieval (pp. 1649-1659).

**Ethical Concerns:**

["NO or VERY MINOR ethics concerns only"]

**Final Justification:**

After reading the rebuttal and discussions, I keep my score and recommend borderline accept because of some unresolved limitations, despite there are some contributions in the proved theory and experiments.

**Limitations:**

In Section 7, the authors have discussed about several limitations of the work, although there is no discussion about negative societal impact.

**Paper Formatting Concerns:**

N/A.

**Quality:**

3

**Strengths And Weaknesses:**

### Strengths
1. IIPM serves a novel framework to systematically apply Choquet integrals to compare capacities in imprecise probability.
2. Robust theoretical basis and creditability of IIPM
3. Empirical experiments on benchmark datasets demonstrate the effectiveness of MMI based on IIPM

### Weaknesses
1. Lack of sampling-based estimation
2. Experiments limited on classification tasks and specific datasets
3. Scalability and efficiency for large-scale problems
4. Lack of experiments of further utilizing uncertainty measures to improve model peformance

---

> ### Author Rebuttal · Authors · 2025-07-27
>
> We thank the reviewer for their insightful and thorough feedback. We will address your questions below:
>
> -----
>
> **Q1: As mentioned in Section 7, a key limitation is the lack of a sampling-based estimation method for IIPMs. Would there be any future plan to address this?**
>
> A1: Yes, we are working on extensions of this work to overcome this problem. One natural direction we are trying out is to utilise IIPM to find the closest precise probability model to a given imprecise probability, and perform sampling from the precise model. Of course, understanding the estimation quality, in this case, is highly non-trivial; therefore, we believe it is outside the scope of this work.
>
> -----
>
> **Q2: What is the trade-off and the key of using the efficient upper bound (or MMI-Lin)? It would be great to have some elaboration.**
>
> A2: In lines 332-336, we provided a sketch proof of Proposition 21. We are happy to clarify it further in the camera-ready version.
> In terms of trade-off, we have performed an ablation study in the appendix (Figure 3), which empirically looked at the difference between the actual MMI and MMI-Lin. While there are some gaps between these numbers, in our selective classification experiments, we could not find any performance difference.
>
> -----
>
> **Q3: Would the proposed framework be able to be applied to other ML tasks like regression and ranking? If so, it would also be great to have more experiments to validate.**
>
> A3: As mentioned in our limitation section, extending it to problems such as regression would require developing further computational tools to evaluate the Choquet integral in a real line. Given that the goal of the current paper is to establish a strong mathematical groundwork for the metric and to justify its relevance to the ML community through the EU quantification problem, we decided to leave the computational development for the regression problem as future work.
>
> ----
>
> **Q4: There are some studies like [a] to leverage uncertainty measures to improve the model quality. Can the proposed framework also do that? It would also be great to have some studies and discussions about this direction.**
>
> A4: We thank the reviewer for pointing this reference out. We will add a brief discussion on this, including the suggested reference, in the camera-ready version.
>
> ----
>
> We hope our responses have added clarity regarding the motivation and applicability of our current work. We kindly ask the reviewer to possibly reconsider their score in light of our discussion. Thank you for your consideration.

---

> > ### Comment · Reviewer_ph2P · 2025-08-05
> >
> > I appreciate authors' responses to my comments. Due to the minor limitations remaining unresolved, I keep my rating as  borderline accept on the positive side for this paper.

---

### Official Review · Reviewer_RsrW · 2025-07-03

**Clarity:** 3
**Significance:** 3
**Originality:** 4
**Rating:** 5
**Confidence:** 3

**Summary:**

This paper highlights the need for a metric to compare imprecise probabilities and introduce a metric called Integral Imprecise Probability Metric (IIPM) framework, a Choquet integral-based generalisation of classical Integral Probability Metrics (IPMs) to the setting of capacities. Based on IIPM, a new epistemic uncertainty measure, Maximum Mean Imprecisions, is also introduced. The theoretical conditions for IIPM are established while MMI is also empirically validated.

**Questions:**

See weaknesses

**Ethical Concerns:**

["NO or VERY MINOR ethics concerns only"]

**Final Justification:**

Some of my concerns about experimental validation were not addressed but I maintain my score.

**Limitations:**

yes

**Paper Formatting Concerns:**

1.	Missing closing brackets in eq. 1
2.	Line 269: “1=P(X)”?

**Quality:**

3

**Strengths And Weaknesses:**

Strengths:
-	The paper comes with strong mathematical setting and presentation of the proposed metric.
-	The metric shows its generalized nature and how it devolves into other popular metrics.

Weaknesses:
-	The paper lacks in experimental evaluation. A widely used way to evaluate epistemic uncertainty is by Out-Of-Distribution (OOD) detection. It might be interesting to see how MMI performs there.
-	The metric requires ensembles of models to compute lower probabilities. This, by default, makes it more resource and time expensive. How does this compare to other available metrics?
-	What effect does the ensemble size have on metric’s performance?
-	It might be nice to add some visual examples to show the efficacy of the metric and what kind of numbers it produces on different inputs.

---

> ### Author Rebuttal · Authors · 2025-07-27
>
> We thank the reviewer for pointing out the typos and for their valuable suggestions. We agree that exploring the interplay between ensemble size and the quantified uncertainty can provide further insight, and we will include an ablation study on this aspect in the camera-ready version.
>
> From a theoretical standpoint, we already know that as we increase the size of the ensemble while keeping the rest of the elements the same, the resulting credal set could only expand or remain unchanged. This implies that the lower probability becomes more conservative (i.e. smaller or unchanged), and by the monotonicity axiom (A3), the quantified EU will increase or stay the same. This property aligns with the intuition that adding diverse hypotheses should reflect greater epistemic caution.
>
> We believe this ablation will help clarify further. Thank you for your suggestion.

---

### Official Review · Reviewer_HsWH · 2025-07-05

**Clarity:** 3
**Significance:** 2
**Originality:** 3
**Rating:** 5
**Confidence:** 2

**Summary:**

The paper discusses IIPMs, a generalization of IPMs that operate on capacities (a class that include non-additive measures). This is a theoretical paper that provides for a sound basis for properties such as (a) suitably chosen IIPMs are proper metrics, (b) new/generalized distances such as Lower Dudley and Lower TV, (c) re-derive a optimal transport result, (d) Maximum Mean Imprecision (MMI, a measure of epistemic uncertainty/EU which also is validated axiomatically) etc. Empirical evidence is also provided to compare with traditional uncertainty quantification measures. A linear time upper bound MMI-Lin is also proposed here.

**Questions:**

Is it possible to get one dataset where EU ground truth is available (semi=synthetic) so that the uncertainty measures can be directly compared?

**Ethical Concerns:**

["NO or VERY MINOR ethics concerns only"]

**Final Justification:**

The responses to concerns raised in the review were satisfactory.

**Limitations:**

yes

**Quality:**

3

**Strengths And Weaknesses:**

The authors provide a principled investigation and derivation of key results for IIPMs and placing them very nicely relative to classical IPMs.
The introduction of MMI as a Epistemic Uncertainty quantification measure is interesting (including its linear-time compute variant)
The paper is well motivated.
In terms of weaknesses, practicality in real world ML settings is not fully fleshed out (understandably so given the page limits). MMI-Lin is a great starting point, but not addressing these practical gaps (e.g., how to use in regression/structured prediction etc) might limit their value to theoretical interest only.

---

> ### Author Rebuttal · Authors · 2025-07-27
>
> We thank the reviewer for their insightful feedback.
>
> **Q1: “In terms of weaknesses, practicality in real-world ML settings is not fully fleshed out (understandably so given the page limits). MMI-Lin is a great starting point, but not addressing these practical gaps (e.g., how to use in regression/structured prediction etc) might limit their value to theoretical interest only.”**
>
> A1: We appreciate the reviewer’s thoughtful comment and fully agree that extending the MMI framework to regression and structured prediction tasks would further broaden its practical impact. Indeed, we are actively investigating these directions beyond the current submission.
>
> Let us emphasise that the theory presented in Section 3 is not limited to discrete spaces—it extends naturally to continuous state spaces, as demonstrated via our use of optimal transport and the epsilon-contaminated MMD framework in Appendix D.2.
>
> Since adapting MMI to regression or structured prediction problems would require the development of additional computational tools and problem-specific modeling choices, we defer it to future work.  Nonetheless, we see this as a natural and exciting avenue for future work, and will clarify the practical use in regression/structured prediction in the camera-ready version.
>
> -----
> **Q2: “Is it possible to get one dataset where EU ground truth is available (semi=synthetic) so that the uncertainty measures can be directly compared?”**
>
> A2: We thank the reviewer for this insightful suggestion. We would like to highlight a fundamental limitation in the evaluation of epistemic uncertainty (EU) that has been established in recent literature. In particular, Bengs et al. [1] show that it is provably impossible to construct a proper scoring rule (i.e., a loss function) that allows for the direct supervision or learning of second-order uncertainty from zeroth-order data (i.e., datasets consisting only of input-output pairs). This result underscores a key epistemological challenge: epistemic uncertainty is inherently about the limitations of knowledge, and not directly observable or labelable from standard supervised data.
>
> As a result, the epistemic uncertainty quantification (UQ) community has largely moved away from using “ground-truth”-based or accuracy-style metrics, which may be appropriate for aleatoric uncertainty but are not well-justified for epistemic UQ. Instead, the standard approach is to evaluate epistemic UQ methods by their utility in downstream tasks where uncertainty matters—such as selective classification, robust decision-making, or exploration—rather than by ground-truth alignment.
>
> In line with this principle, our evaluation focuses on selective classification as a practical proxy for assessing the value of uncertainty estimates in real-world scenarios. We believe this reflects a more meaningful and principled evaluation of epistemic UQ methods.
>
> Reference:
> [1] Bengs, Viktor, Eyke Hüllermeier, and Willem Waegeman. “On second-order scoring rules for epistemic uncertainty quantification.” International Conference on Machine Learning. PMLR, 2023.
>
> ---------
>
> We hope our responses have added clarity regarding the motivation and applicability of our current work. We kindly ask the reviewer to possibly reconsider their score in light of our discussion. Thank you for your consideration.

---

### Note · Authors · 2025-08-11

## Our contributions.
We propose the framework of integral imprecise probability metrics (IIPM), a generalisation of the widely used integral probability metrics (IPM) to imprecise probabilities. In Section 3, we provide a rigorous analysis, establishing when IIPM defines a valid metric, how it extends classical metrics, and how it recovers key results from prior work.
In Section 4, we show that measuring the divergence between an imprecise probability measure and its conjugate via IIPM yields a novel epistemic uncertainty quantification function, maximum mean imprecision (MMI). MMI satisfies standard UQ axioms and, for a specific function class, admits a tight upper bound computable in linear time—making it far more practical than other axiomatic approaches, which often require exponential computation. We validate our method on selective classification, achieving state-of-the-art results, especially for large-class problems. We believe IIPM lays a solid foundation for future developments in epistemic UQ, much like IPM has for many modern algorithms.

## Reviewer concerns and responses.
(a) On empirical scope (the paper does not have extensive experiments such as OOD detection):
- Given space and scope constraints, we prioritised axiomatic justification over additional empirical evaluation. In epistemic UQ, due to lack of ground truth, axiomatic approach is deemed more reasonable than indirect downstream tasks, which can only falsify poor methods rather than confirm good ones.

(b) On relevance (imprecise probabilities might not be too relevant to the ML community):
- We identified at least 16 papers in top ML venues (ICLR, ICML, NeurIPS, past 3 years) that use imprecise probabilities as a core methodology—indicating growing adoption in trustworthy ML research.

(c) On applicability (this paper only studies classification problem, aka on discrete state space, what about regression or ranking problems?):
- While regression UQ remains an open and technically demanding problem, in our Section 3, the IIPM results apply broadly to both discrete and continuous spaces. We will extend our framework to regression and other problems but it would require extensive effort and thus is outside of the scope of this project.

---

### Decision · Program_Chairs · 2025-09-17

**Decision:**

Accept (poster)

**Comment:**

- Summary: This paper proposes Integral Imprecise Probability Metric framework, which is an extension of integral probability metrics to operate on capacities, instead of probability measures. It establishes the conditions when the proposed metrics are valid. The authors also propose Maximum Mean Imprecisions to enable epistemic uncertainty quantification.
- Strengths:
  - The framework is motivated by robust theoretical analysis.
  - The proposed method is novel and the paper is well written and motivated.
- Weaknesses:
  - The application of the approach to practical settings remains unexplored in addition to its scalability to large scale ML problems.
  - The relevance of the problem to the Neurips community is highlighted as a potential limitation.
  - Empirical evaluations can be stronger.
    - Reviewer HsWH suggested comparisons with respect to simulated settings where empirical uncertainty is controlled. Authors provided their motivation on why such experiments are not ideal. Even though I acknowledge that there are limitations, I believe the suggested experiments remain useful. Similarly, measuring utility in downstream tasks might be also subject to limitations.
    - Reviewer RsrW suggested OOD detection, which is not addressed during the rebuttal phase by the authors.
- Suggestions:
  - I think focusing on the empirical evaluations and expanding the experiments to cover more downstream tasks as well as simulated settings would make this paper stronger and help with its wider adoption.
- Recommendation:
  - All of the four reviewers recommended acceptance of the paper, two of whom gave borderline acceptance scores.
  - I believe this paper will be a good contribution to Neurips 2025. The paper tackles uncertainty quantification, which is a significant research problem. All of the reviewers acknowledge the strong theoretical foundations of the proposed approach.